# LEARNING TO INFER

## ABSTRACT

Inference models, which replace an optimization-based inference procedure with a learned model, have been fundamental in advancing Bayesian deep learning, the most notable example being variational auto-encoders (VAEs). In this paper, we propose *iterative inference models*, which learn how to optimize a variational lower bound through repeatedly encoding gradients. Our approach generalizes VAEs under certain conditions, and by viewing VAEs in the context of iterative inference, we provide further insight into several recent empirical findings. We demonstrate the inference optimization capabilities of iterative inference models, explore unique aspects of these models, and show that they outperform standard inference models on typical benchmark data sets.

## 1 INTRODUCTION

Generative models present the possibility of learning structure from data in unsupervised or semi-supervised settings, thereby facilitating more flexible systems to learn and perform tasks in computer vision, robotics, and other application domains with limited human involvement. Latent variable models, a class of generative models, are particularly well-suited to learning hidden structure. They frame the process of data generation as a mapping from a set of latent variables underlying the data. When this mapping is parameterized by a deep neural network, the model can learn complex, non-linear relationships, such as object identities (Higgins et al. (2016)) and dynamics (Xue et al. (2016); Karl et al. (2017)). However, performing exact posterior inference in these models is computationally intractable, necessitating the use of approximate inference methods.

Variational inference (Hinton & Van Camp (1993); Jordan et al. (1998)) is a scalable approximate inference method, transforming inference into a non-convex optimization problem. Using a set of approximate posterior distributions, e.g. Gaussians, variational inference attempts to find the distribution that most closely matches the true posterior. This matching is accomplished by maximizing a lower bound on the marginal log-likelihood, or model evidence, which can also be used to learn the model parameters. The ensuing expectation-maximization procedure alternates between optimizing the approximate posteriors and model parameters (Dempster et al. (1977); Neal & Hinton (1998); Hoffman et al. (2013)). Amortized inference (Gershman & Goodman (2014)) avoids exactly computing optimized approximate posterior distributions for each data example, instead learning a separate inference model to perform this task. Taking the data example as input, this model outputs an estimate of the corresponding approximate posterior. When the generative and inference models are parameterized with neural networks, the resulting set-up is referred to as a variational auto-encoder (VAE) (Kingma & Welling (2014); Rezende et al. (2014)).

We introduce a new class of inference models, referred to as iterative inference models, inspired by recent work in learning to learn (Andrychowicz et al. (2016)). Rather than directly mapping the data to the approximate posterior, these models learn how to iteratively estimate the approximate posterior by repeatedly encoding the corresponding gradients, i.e. learning to infer. With inference computation distributed over multiple iterations, we conjecture that this model set-up should provide improved inference estimates over standard inference models given sufficient model capacity. Our work is presented as follows: Section 2 contains background on latent variable models, variational inference, and inference models; Section 3 motivates and introduces iterative inference models; Section 4 presents this approach for latent Gaussian models, showing that a particular form of iterative inference models reduces to standard inference models under mild assumptions; Section 5 contains empirical results; and Section 6 concludes our work.

## 2 BACKGROUND

### 2.1 LATENT VARIABLE MODELS & VARIATIONAL INFERENCE

Latent variable models are generative probabilistic models that use local (per data example) latent variables, $\mathbf{z}$, to model observations, $\mathbf{x}$, using global (across data examples) parameters, $\theta$. A model is defined by the joint distribution $p_\theta(\mathbf{x}, \mathbf{z}) = p_\theta(\mathbf{x}|\mathbf{z})p_\theta(\mathbf{z})$, which is composed of the conditional likelihood and the prior. Learning the model parameters and inferring the posterior $p(\mathbf{z}|\mathbf{x})$ are intractable for all but the simplest models, as they require evaluating the marginal likelihood, $p_\theta(\mathbf{x}) = \int p_\theta(\mathbf{x}, \mathbf{z})d\mathbf{z}$, which involves integrating the model over $\mathbf{z}$. For this reason, we often turn to approximate inference methods.

Variational inference reformulates this intractable integration as an optimization problem by introducing an approximate posterior[1] $q(\mathbf{z}|\mathbf{x})$, typically chosen from some tractable family of distributions, and minimizing the KL-divergence from the true posterior, $D_{KL}(q(\mathbf{z}|\mathbf{x})||p(\mathbf{z}|\mathbf{x}))$. This quantity cannot be minimized directly, as it contains the true posterior. Instead, the KL-divergence can be decomposed into

$$D_{KL}(q(\mathbf{z}|\mathbf{x})||p(\mathbf{z}|\mathbf{x})) = \log p_\theta(\mathbf{x}) - \mathcal{L}, \tag{1}$$

where $\mathcal{L}$ is the evidence lower bound (ELBO), which is defined as:

$$\mathcal{L} \equiv \mathbb{E}_{\mathbf{z} \sim q(\mathbf{z}|\mathbf{x})}\left[\log p_\theta(\mathbf{x}, \mathbf{z}) - \log q(\mathbf{z}|\mathbf{x})\right] \tag{2}$$

$$= \mathbb{E}_{\mathbf{z} \sim q(\mathbf{z}|\mathbf{x})}\left[\log p_\theta(\mathbf{x}|\mathbf{z})\right] - D_{KL}(q(\mathbf{z}|\mathbf{x})||p_\theta(\mathbf{z})). \tag{3}$$

Briefly, the first term in eq. 3 can be considered as a reconstruction term, as it expresses how well the output fits the data example. The second term can be considered as a regularization term, as it quantifies the dissimilarity between the latent representation and the prior. Because $\log p_\theta(\mathbf{x})$ is not a function of $q(\mathbf{z}|\mathbf{x})$, in eq. 1 we can minimize $D_{KL}(q(\mathbf{z}|\mathbf{x})||p(\mathbf{z}|\mathbf{x}))$, thereby performing approximate *inference*, by maximizing $\mathcal{L}$ w.r.t. $q(\mathbf{z}|\mathbf{x})$. Likewise, because $D_{KL}(q(\mathbf{z}|\mathbf{x})||p(\mathbf{z}|\mathbf{x}))$ is non-negative, $\mathcal{L}$ is a lower bound on $\log p_\theta(\mathbf{x})$, meaning that if we have inferred an optimal $q(\mathbf{z}|\mathbf{x})$, *learning* corresponds to maximizing $\mathcal{L}$ w.r.t. $\theta$.

### 2.2 VARIATIONAL EXPECTATION MAXIMIZATION (EM)

The optimization procedures involved in inference and learning, when implemented using conventional gradient ascent techniques, are respectively the expectation and maximization steps of the variational EM algorithm (Dempster et al. (1977); Neal & Hinton (1998); Hoffman et al. (2013)), which alternate until convergence. When $q(\mathbf{z}|\mathbf{x})$ takes a parametric form, the expectation step for data example $\mathbf{x}^{(i)}$ involves finding a set of distribution parameters, $\boldsymbol{\lambda}^{(i)}$, that are optimal. With a factorized Gaussian density over continuous variables, i.e. $\boldsymbol{\lambda}^{(i)} = \{\boldsymbol{\mu}_q^{(i)}, \boldsymbol{\sigma}_q^{2(i)}\}$ and $q(\mathbf{z}^{(i)}|\mathbf{x}^{(i)}) = \mathcal{N}(\mathbf{z}^{(i)}; \boldsymbol{\mu}_q^{(i)}, \operatorname{diag} \boldsymbol{\sigma}_q^{2(i)})$, this entails repeatedly estimating the stochastic gradients $\nabla_{\boldsymbol{\lambda}^{(i)}}\mathcal{L}$ to optimize $\mathcal{L}$ w.r.t. $\boldsymbol{\lambda}^{(i)}$. This direct optimization procedure, which is repeated for each example, is not only computationally costly for expressive generative models and large data sets, but also sensitive to step sizes and initial conditions.

### 2.3 INFERENCE MODELS

Amortized inference (Gershman & Goodman (2014)) replaces the optimization of each set of local approximate posterior parameters, $\boldsymbol{\lambda}^{(i)}$, with the optimization of a set of global parameters, $\phi$, contained within an inference model. Taking $\mathbf{x}^{(i)}$ as input, this model directly outputs estimates of $\boldsymbol{\lambda}^{(i)}$. Sharing the inference model across data examples allows for an efficient algorithm, in which $\phi$ and $\theta$ can be updated jointly. The canonical example, the variational auto-encoder (VAE) (Kingma & Welling (2014); Rezende et al. (2014)), employs the reparameterization trick to propagate stochastic gradients from the generative model to the inference model, both of which are parameterized by neural networks. The formulation has an intuitive interpretation: the inference model *encodes* $\mathbf{x}$ into $q(\mathbf{z}|\mathbf{x})$, and the generative model *decodes* samples from $q(\mathbf{z}|\mathbf{x})$ into $p(\mathbf{x}|\mathbf{z})$. Throughout the rest of this paper, we refer to inference models of this form as *standard inference models*.

---

[1] We use $q(\mathbf{z}|\mathbf{x})$ to denote that the approximate posterior is conditioned on a data example (i.e. local), however this need not be through a direct functional dependence.

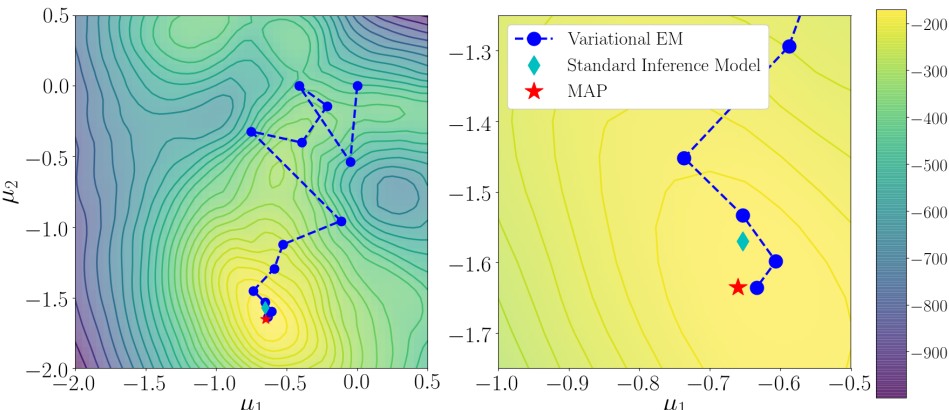

Figure 1: Optimization surface of $\mathcal{L}$ (in *nats*) for a 2-D latent Gaussian model and a particular MNIST data example. Shown on the plot are the MAP (optimal estimate), the output of a standard inference model (VAE), and an expectation step trajectory of variational EM using stochastic gradient ascent. The plot on the right shows the estimates of each inference scheme near the optimum. The expectation step arrives at a better final inference estimate than the standard inference model.

## 3 ITERATIVE INFERENCE MODELS

In Section 3.2, we introduce our contribution, iterative inference models. We first motivate our approach in Section 3.1 by interpreting standard inference models in VAEs as optimization models, i.e. models that learn to perform optimization. Using insights from other optimization models, this interpretation extends and improves upon standard inference models.

### 3.1 INFERENCE MODELS ARE OPTIMIZATION MODELS

As described in Section 2.1, variational inference transforms inference into the maximization of $\mathcal{L}$ w.r.t. the parameters of $q(\mathbf{z}|\mathbf{x})$, constituting the expectation step of the variational EM algorithm. In general, this is a non-convex optimization problem, making it somewhat surprising that an inference model can *learn* to output reasonable estimates of $q(\mathbf{z}|\mathbf{x})$ across data examples. Of course, directly comparing inference schemes is complicated by the fact that generative models adapt to accommodate their approximate posteriors. Nevertheless, inference models attempt to replace traditional optimization techniques with a learned mapping from $\mathbf{x}$ to $q(\mathbf{z}|\mathbf{x})$.

We demonstrate this point in Figure 1 by visualizing the optimization surface of $\mathcal{L}$ defined by a trained 2-D latent Gaussian model and a particular data example, in this case, a binarized MNIST digit. To visualize the surface, we use a 2-D point estimate as the approximate posterior, $q(\mathbf{z}|\mathbf{x}) = \delta(\mathbf{z} = \boldsymbol{\mu}_q)$, where $\boldsymbol{\mu}_q = (\mu_1, \mu_2) \in \mathbb{R}^2$ and $\delta$ is the Dirac delta function. See Appendix C.1 for further details. Shown on the plot are the MAP (i.e. optimal) estimate, the estimate from a trained inference model, and an expectation step trajectory using stochastic gradient ascent on $\boldsymbol{\mu}_q$. The expectation step arrives at a better final estimate, but it requires many iterations and is dependent on the step size and initial estimate. The inference model outputs a near-optimal estimate in one forward pass without hand tuning (other than the architecture), but it is restricted to this single estimate. Note that the inference model does not attain the optimal estimate, resulting in an *"amortization gap"* (Cremer et al. (2017)).

This example illustrates how inference models differ from conventional optimization techniques. Despite having no convergence guarantees on inference optimization, inference models have been shown to work well empirically. However, by learning a direct mapping from $\mathbf{x}$ to $q(\mathbf{z}|\mathbf{x})$, standard inference models are restricted to only single-step estimation procedures, which may yield worse inference estimates. The resulting large amortization gap then limits the quality of the accompanying generative model. To improve upon this paradigm, we take inspiration from the area of learning to learn, where Andrychowicz et al. (2016) showed that an *optimizer* model, instantiated as a recurrent neural network, can learn to optimize the parameters of an *optimizee* model, another neural network,

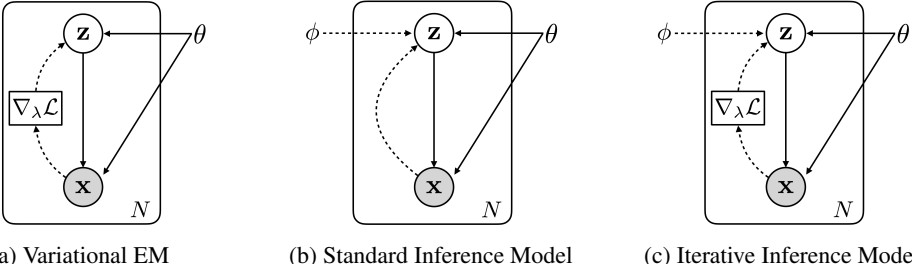

(a) Variational EM          (b) Standard Inference Model          (c) Iterative Inference Model

Figure 2: Plate notation for a latent variable model (solid lines) with each inference scheme (dashed lines). $\theta$ refers to the generative model (decoder) parameters. $\nabla_{\boldsymbol{\lambda}}\mathcal{L}$ denotes the gradients of the ELBO w.r.t. the distribution parameters, $\boldsymbol{\lambda}$, of the approximate posterior, $q(\mathbf{z}|\mathbf{x})$. Iterative inference models learn to perform approximate inference optimization by using these gradients and a set of inference model (encoder) parameters, $\phi$. See Figure 8 for a similar set of diagrams with unrolled computational graphs.

for various tasks. The optimizer model receives the optimizee's parameter gradients and outputs updates to these parameters to improve the optimizee's loss. Because the computational graph is differentiable, the optimizer itself can also be learned. Optimization models can learn to adaptively adjust update step sizes, potentially speeding up and improving optimization.

While Andrychowicz et al. (2016) focus primarily on parameter optimization (i.e. learning), we apply an analogous approach to inference optimization in latent variable models. We refer to this class of optimization models as *iterative inference models*, as they are inference models that iteratively update their approximate posterior estimates. Our work differs from that of Andrychowicz et al. (2016) in three distinct ways: (1) variational inference is a qualitatively different optimization problem, involving amortization across data examples rather than learning tasks; (2) we utilize non-recurrent optimization models, providing a more computationally efficient model that breaks the assumption that previous gradient information is essential for learned optimization; and (3) we provide a novel model formulation that *approximates* gradient steps using locally computed errors on latent and observed variables (see Section 4.1). We formalize our approach in the following section.

## 3.2 ITERATIVE INFERENCE MODELS

We present iterative inference models starting from the context of standard inference models. For a standard inference model $f$ with parameters $\phi$, the estimate of the approximate posterior distribution parameters $\boldsymbol{\lambda}^{(i)}$ for data example $\mathbf{x}^{(i)}$ is of the form:

$$\boldsymbol{\lambda}^{(i)} = f(\mathbf{x}^{(i)}; \phi). \tag{4}$$

We propose to instead use an iterative inference model, also denoted as $f$ with parameters $\phi$. With $\mathcal{L}_t^{(i)} \equiv \mathcal{L}(\mathbf{x}^{(i)}, \boldsymbol{\lambda}_t^{(i)}; \theta)$ as the ELBO for data example $\mathbf{x}^{(i)}$ at inference iteration $t$, the model uses the approximate posterior gradients, denoted $\nabla_{\boldsymbol{\lambda}}\mathcal{L}_t^{(i)}$, to output updated estimates of $\boldsymbol{\lambda}^{(i)}$:

$$\boldsymbol{\lambda}_{t+1}^{(i)} = f_t(\nabla_{\boldsymbol{\lambda}}\mathcal{L}_t^{(i)}, \boldsymbol{\lambda}_t^{(i)}; \phi), \tag{5}$$

where $\boldsymbol{\lambda}_t^{(i)}$ is the estimate of $\boldsymbol{\lambda}^{(i)}$ at inference iteration $t$. We use $f_t$ to highlight that the form of $f$ at iteration $t$ may depend on hidden states within the iterative inference model, such as those found within recurrent neural networks. See Figures 2 and 8 for schematic comparisons of iterative inference models with variational EM and standard inference models. As with standard inference models, the parameters of an iterative inference model can be updated using stochastic estimates of $\nabla_{\phi}\mathcal{L}$, obtained through the reparameterization trick or other methods. Model parameter updating is typically performed using standard optimization techniques. Note that eq. 5 is in a general form and contains, as a special case, the residual updating scheme used in Andrychowicz et al. (2016).

## 4    ITERATIVE INFERENCE IN LATENT GAUSSIAN MODELS

We now describe an example of iterative inference models for latent Gaussian generative models, deriving the gradients to understand the source of the approximate posterior updates. Latent Gaussian models are latent variable models with Gaussian prior distributions over latent variables: $p(\mathbf{z}) = \mathcal{N}(\mathbf{z}; \boldsymbol{\mu}_p, \mathrm{diag}\,\boldsymbol{\sigma}_p^2)$. This class of models is often used in VAEs and is a common choice for representing continuous-valued latent variables. While the approximate posterior can be any probability density, it is typically also chosen as Gaussian: $q(\mathbf{z}|\mathbf{x}) = \mathcal{N}(\mathbf{z}; \boldsymbol{\mu}_q, \mathrm{diag}\,\boldsymbol{\sigma}_q^2)$. With this choice, $\boldsymbol{\lambda}^{(i)}$ corresponds to $\{\boldsymbol{\mu}_q^{(i)}, \boldsymbol{\sigma}_q^{2(i)}\}$ for example $\mathbf{x}^{(i)}$. Dropping the superscript $(i)$ to simplify notation, we can express eq. 5 for this model as:

$$\boldsymbol{\mu}_{q,t+1} = f_t^{\boldsymbol{\mu}_q}(\nabla_{\boldsymbol{\mu}_q}\mathcal{L}_t, \boldsymbol{\mu}_{q,t}; \phi), \tag{6}$$

$$\boldsymbol{\sigma}_{q,t+1}^2 = f_t^{\boldsymbol{\sigma}_q^2}(\nabla_{\boldsymbol{\sigma}_q^2}\mathcal{L}_t, \boldsymbol{\sigma}_{q,t}^2; \phi), \tag{7}$$

where $f_t^{\boldsymbol{\mu}_q}$ and $f_t^{\boldsymbol{\sigma}_q^2}$ are the iterative inference models for updating $\boldsymbol{\mu}_q$ and $\boldsymbol{\sigma}_q^2$ respectively. For continuous observations, we can use a Gaussian output density: $p(\mathbf{x}|\mathbf{z}) = \mathcal{N}(\mathbf{x}; \boldsymbol{\mu}_\mathbf{x}, \mathrm{diag}\,\boldsymbol{\sigma}_\mathbf{x}^2)$. Here, $\boldsymbol{\mu}_\mathbf{x} = \boldsymbol{\mu}_\mathbf{x}(\mathbf{z})$ is a non-linear function of $\mathbf{z}$, and $\boldsymbol{\sigma}_\mathbf{x}^2$ is a global parameter, a common assumption in these models. The approximate posterior parameter gradients for this model are (see Appendix A):

$$\nabla_{\boldsymbol{\mu}_q}\mathcal{L} = \mathbb{E}_{\mathcal{N}(\boldsymbol{\epsilon};\mathbf{0},\mathbf{I})}\left[\frac{\partial\boldsymbol{\mu}_\mathbf{x}}{\partial\boldsymbol{\mu}_q}^\mathsf{T}\frac{\mathbf{x} - \boldsymbol{\mu}_\mathbf{x}}{\boldsymbol{\sigma}_\mathbf{x}^2} - \frac{\boldsymbol{\mu}_q + \boldsymbol{\sigma}_q \odot \boldsymbol{\epsilon} - \boldsymbol{\mu}_p}{\boldsymbol{\sigma}_p^2}\right] \tag{8}$$

$$\nabla_{\boldsymbol{\sigma}_q^2}\mathcal{L} = \mathbb{E}_{\mathcal{N}(\boldsymbol{\epsilon};\mathbf{0},\mathbf{I})}\left[\frac{\partial\boldsymbol{\mu}_\mathbf{x}}{\partial\boldsymbol{\sigma}_q^2}^\mathsf{T}\frac{\mathbf{x} - \boldsymbol{\mu}_\mathbf{x}}{\boldsymbol{\sigma}_\mathbf{x}^2} - \left(\mathrm{diag}\,\frac{\boldsymbol{\epsilon}}{2\boldsymbol{\sigma}_q}\right)^\mathsf{T}\frac{\boldsymbol{\mu}_q + \boldsymbol{\sigma}_q \odot \boldsymbol{\epsilon} - \boldsymbol{\mu}_p}{\boldsymbol{\sigma}_p^2}\right] - \frac{\mathbf{1}}{2\boldsymbol{\sigma}_q^2}, \tag{9}$$

where $\boldsymbol{\epsilon} \sim \mathcal{N}(\mathbf{0},\mathbf{I})$ is the auxiliary noise variable from the reparameterization trick, $\odot$ denotes element-wise multiplication, and all division is performed element-wise. In Appendix A, we also derive the corresponding gradients for a Bernoulli output distribution, which take a similar form. Although we only derive gradients for these two output distributions, note that iterative inference models can be used with *any* distribution form. We now briefly discuss the terms in eqs. 8 and 9. Re-expressing the reparameterized latent variable as $\mathbf{z} = \boldsymbol{\mu}_q + \boldsymbol{\sigma}_q \odot \boldsymbol{\epsilon}$, the gradients have two shared terms, $(\mathbf{x} - \boldsymbol{\mu}_\mathbf{x})/\boldsymbol{\sigma}_\mathbf{x}^2$ and $(\mathbf{z} - \boldsymbol{\mu}_p)/\boldsymbol{\sigma}_p^2$, the precision-weighted errors at the observed ("bottom-up") and latent ("top-down") levels respectively. The terms $\frac{\partial\boldsymbol{\mu}_\mathbf{x}}{\partial\boldsymbol{\mu}_q}$ and $\frac{\partial\boldsymbol{\mu}_\mathbf{x}}{\partial\boldsymbol{\sigma}_q^2}$ are the Jacobian matrices of $\boldsymbol{\mu}_\mathbf{x}$ w.r.t. the approximate posterior parameters, which effectively *invert* the output model. Understanding the significance of each term, in the following section we provide an alternative formulation of iterative inference models for latent Gaussian generative models.

### 4.1    APPROXIMATING THE APPROXIMATE POSTERIOR GRADIENTS

The approximate posterior gradients are inherently stochastic, arising from the fact that evaluating $\mathcal{L}$ involves approximating expectations (eq. 2) using Monte Carlo samples of $\mathbf{z} \sim q(\mathbf{z}|\mathbf{x})$. As these estimates always contain some degree of noise, a close *approximation* to these gradients should also suffice for updating the approximate posterior parameters. The motivations for this are two-fold: (1) approximate gradients may be easier to compute, especially in an online setting, and (2) by encoding more general terms, the inference model may be able to approximate higher-order approximate posterior derivatives, allowing for faster convergence. We now provide an alternative formulation of iterative inference models for latent Gaussian models that *approximates* gradient information.

With the exception of $\frac{\partial\boldsymbol{\mu}_\mathbf{x}}{\partial\boldsymbol{\mu}_q}$ and $\frac{\partial\boldsymbol{\mu}_\mathbf{x}}{\partial\boldsymbol{\sigma}_q^2}$, all terms in eqs. 8 and 9 can be easily computed using $\mathbf{x}$ and the distribution parameters of $p(\mathbf{x}|\mathbf{z})$, $p(\mathbf{z})$, and $q(\mathbf{z}|\mathbf{x})$. Likewise, higher-order approximate posterior derivatives consist of these common terms as well as higher-order derivatives of the output model. As the output model derivatives are themselves *functions*, by encoding only the common terms, we can offload these (approximate) derivative calculations onto the iterative inference model. Again dropping the superscript $(i)$, one possible set-up is formulated as follows:

$$\boldsymbol{\mu}_{q,t+1} = f_t^{\boldsymbol{\mu}_q}(\boldsymbol{\varepsilon}_{\mathbf{x},t}, \boldsymbol{\varepsilon}_{\mathbf{z},t}, \boldsymbol{\mu}_{q,t}; \phi), \tag{10}$$

$$\boldsymbol{\sigma}_{q,t+1}^2 = f_t^{\boldsymbol{\sigma}_q^2}(\boldsymbol{\varepsilon}_{\mathbf{x},t}, \boldsymbol{\varepsilon}_{\mathbf{z},t}, \boldsymbol{\sigma}_{q,t}^2; \phi), \qquad (11)$$

where, in the case of a Gaussian output density, the stochastic error terms are defined as

$$\boldsymbol{\varepsilon}_{\mathbf{x},t} \equiv \mathbb{E}_{\boldsymbol{\epsilon}_t}[(\mathbf{x} - \boldsymbol{\mu}_{t,\mathbf{x}})/\boldsymbol{\sigma}_{\mathbf{x}}^2], \qquad \boldsymbol{\varepsilon}_{\mathbf{z},t} \equiv \mathbb{E}_{\boldsymbol{\epsilon}_t}[(\boldsymbol{\mu}_{q,t} + \boldsymbol{\sigma}_{q,t} \odot \boldsymbol{\epsilon}_t - \boldsymbol{\mu}_p)/\boldsymbol{\sigma}_p^2].$$

This encoding scheme resembles the approach taken in DRAW (Gregor et al. (2015)), where reconstruction errors, $\mathbf{x} - \boldsymbol{\mu}_{t,\mathbf{x}}$, are iteratively encoded. However, DRAW and later variants (Gregor et al. (2016)) do not explicitly account for latent errors, $\boldsymbol{\varepsilon}_{\mathbf{z},t}$, or approximate posterior estimates. If possible, these terms must instead be implicitly handled by the inference model's hidden states. In Section 5.2, we demonstrate that iterative inference models of this form do indeed learn to infer. Unlike gradient encoding iterative inference models, these error encoding models do not require gradients at test time and they empirically perform well even with few inference iterations.

### 4.2 RELATIONSHIP TO CONVENTIONAL VARIATIONAL AUTO-ENCODERS

Under a certain set of assumptions, single-iteration iterative inference models of the derivative approximating form proposed in Section 4.1 are equivalent to standard inference models, as used in conventional VAEs. Specifically, assuming:

1. the initial approximate posterior estimate is a global constant: $\mathcal{N}(\mathbf{z}; \boldsymbol{\mu}_{q,0}, \operatorname{diag}\boldsymbol{\sigma}_{q,0}^2)$,
2. the prior is a global constant: $\mathcal{N}(\mathbf{z}; \boldsymbol{\mu}_p, \operatorname{diag}\boldsymbol{\sigma}_p^2)$,
3. we are in the limit of infinite samples of the initial auxiliary variable $\boldsymbol{\epsilon}_0$,

then the initial approximate posterior estimate $(\boldsymbol{\mu}_{q,0}, \boldsymbol{\sigma}_{q,0}^2)$ and initial latent error $(\boldsymbol{\varepsilon}_{\mathbf{z},0})$ are constants and the initial observation error $(\boldsymbol{\varepsilon}_{\mathbf{x},0})$ is a constant affine transformation of the observation $(\mathbf{x})$. When the inference model is a neural network, then encoding $\mathbf{x}$ or an affine transformation of $\mathbf{x}$ is equivalent (assuming the inputs are properly normalized). Therefore, eqs. 10 and 11 simplify to that of a standard inference model, eq. 4. From this perspective, standard inference models can be interpreted as single-step optimization models that learn to approximate derivatives at a single latent point. In the following section, we consider the case in which the second assumption is violated; iterative inference models naturally handle this case, whereas standard inference models do not.

### 4.3 EXTENSION: INFERENCE IN HIERARCHICAL LATENT VARIABLE MODELS

Hierarchical latent variable models contain higher level latent variables that provide *empirical priors* on lower level variables; $p_\theta(\mathbf{z})$ is thus observation-dependent (see Figure 7 in Appendix A.6). The approximate posterior gradients for an intermediate level in a hierarchical latent Gaussian model (see Appendix A.6) take a similar form as eqs. 8 and 9, comprising bottom-up errors from lower variables and top-down errors from higher variables. Iterative inference models encode both of these errors, either directly or through the gradient. However, standard inference models, which map $\mathbf{x}$ and lower latent variables to each level of latent variables, can only approximate bottom-up information. Lacking top-down prior information, these models must either use a less expressive prior or output poor approximate posterior estimates. Sønderby et al. (2016) identified this phenomenon, proposing a "top-down inference" technique. Iterative inference models formalize and extend this technique.

## 5 EXPERIMENTS

We performed experiments using latent Gaussian models trained on MNIST, Omniglot (Lake et al. (2013)), Street View House Numbers (Netzer et al. (2011)), and CIFAR-10 (Krizhevsky & Hinton (2009)). MNIST and Omniglot were dynamically binarized and modeled with Bernoulli output distributions, and Street View House Numbers and CIFAR-10 were modeled with Gaussian output distributions, using the procedure from Gregor et al. (2016). All experiments presented here use fully-connected neural networks. Reported values of $\mathcal{L}$ were estimated using 1 sample (Figures 3, 5, 6), and reported values of $-\log p(\mathbf{x})$ were estimated using 5,000 importance weighted samples (Table 1). Additional experiment details, including model architectures and optimizers, can be found in Appendix C. We present additional experiments on text data in Appendix D. Source code will be released online.

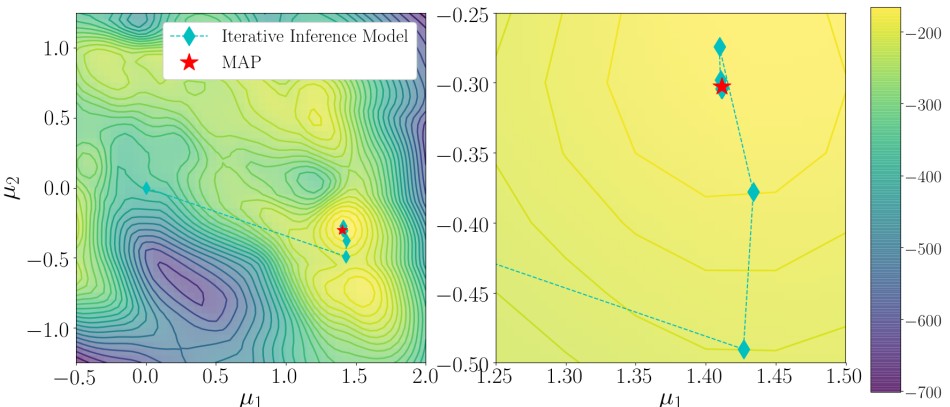

Figure 3: Optimization trajectory along $\mathcal{L}$ (in *nats*) of an iterative inference model with a 2D latent Gaussian model for a particular MNIST test example. The iterative inference model learns to adaptively adjust inference update step sizes to iteratively refine the approximate posterior estimate.

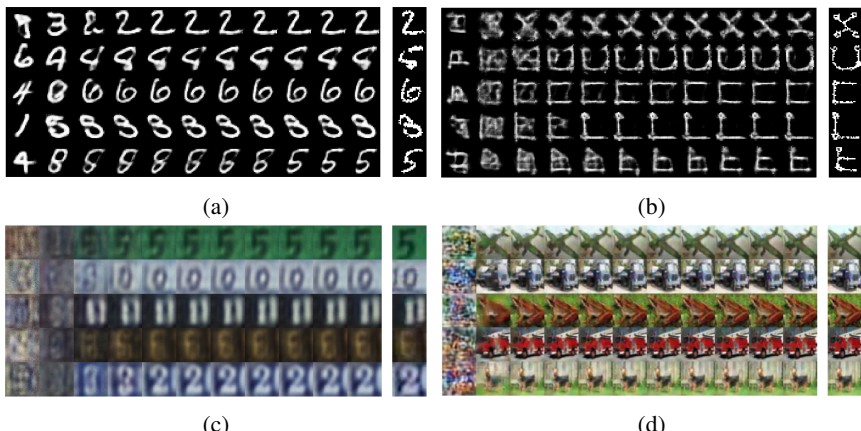

Figure 4: Reconstructions over inference iterations (left to right) for test examples from **(a)** MNIST, **(b)** Omniglot, **(c)** Street View House Numbers, and **(d)** CIFAR-10. Corresponding data examples are shown on the far right of each panel. Empirically, reconstructions become gradually sharper as the iterative inference models traverse the optimization surface, remaining stable after many iterations.

## 5.1 VISUALIZING APPROXIMATE INFERENCE OPTIMIZATION

To confirm the ability of iterative inference models to optimize the approximate posterior, we tested these models in the simplified setting of a 2D latent Gaussian model, trained on MNIST, with a point estimate approximate posterior. The generative model architecture and approximate posterior form are identical to those used in Section 3.1 (see Appendix C.1). Here we show a result from encoding $\mathbf{x}$ and $\nabla_{\boldsymbol{\mu}_q}\mathcal{L}$ through a feedforward neural network. In Figure 3, we visualize an optimization trajectory taken by this model for a particular test example. Despite lacking convergence guarantees, the model learns to adaptively adjust inference update step sizes to navigate the optimization surface, arriving and remaining at a near-optimal approximate posterior estimate for this example.

Approximate inference optimization can also be visualized through data reconstructions. In eq. 3, the reconstruction term encourages $q(\mathbf{z}|\mathbf{x})$ to represent outputs that closely match the data examples. As this is typically the dominant term in $\mathcal{L}$, during inference optimization, the output reconstructions should improve in terms of visual quality, more closely matching $\mathbf{x}$. We demonstrate this phenomenon with iterative inference models for several data sets in Figure 4 (see Appendix C.2 for additional reconstructions.). Reconstruction quality noticeably improves during inference.

## 5.2 Additional Latent Samples & Inference Iterations

We highlight two unique aspects of iterative inference models: direct improvement with additional samples and inference iterations. These aspects provide two advantageous qualitative differences over standard inference models. Additional approximate posterior samples provide more precise gradient estimates, potentially allowing an iterative inference model to output more precise updates. To verify this, we trained standard and iterative inference models on MNIST using 1, 5, 10, and 20 approximate posterior samples. Iterative inference models were trained by encoding the data ($\mathbf{x}$) and approximate posterior gradients ($\nabla_{\boldsymbol{\lambda}}\mathcal{L}$) for 5 iterations. The results are shown in Figure 5a, where we observe that the iterative inference model improves by more than 1 nat with additional samples, while the standard inference model improves by roughly 0.5 nats.

We investigated the effect of training with additional inference iterations while encoding approximate posterior gradients ($\nabla_{\boldsymbol{\lambda}}\mathcal{L}$) or errors ($\boldsymbol{\varepsilon}_{\mathbf{x}}, \boldsymbol{\varepsilon}_{\mathbf{z}}$), with or without the data ($\mathbf{x}$). Section 4 and Appendix A define these terms. Note that the encoded terms affect the number of input parameters to the inference model. Here, the iterative inference model that only encodes $\nabla_{\boldsymbol{\lambda}}\mathcal{L}$ has fewer input parameters than a standard inference model, whereas the models that encode errors or data have strictly more input parameters. Experiments were performed on MNIST, with results for 2, 5, 10, and 16 inference iterations in Figure 5b. All encoding schemes outperformed standard inference models with the same architecture, which we found to be consistent over a range of architectures. Encoding the data was beneficial, allowing the inference model to trade off between learning a direct and iterative mapping. Encoding errors allows the iterative inference model to approximate higher order derivatives (Section 4.1), which we observe helps when training with fewer inference iterations. However, it appears that these approximations are less helpful with additional iterations, where derivative approximation errors likely limit performance.

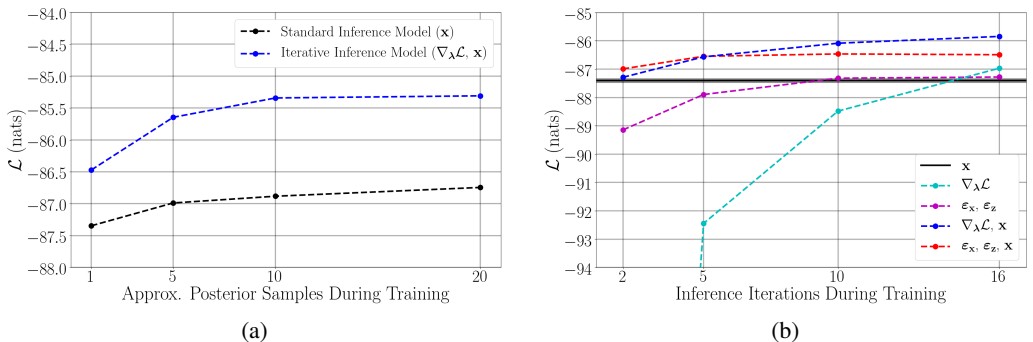

(a)  (b)

Figure 5: Test performance on MNIST of standard and iterative inference models for **(a)** additional samples and **(b)** additional inference iterations during training. Iterative inference models improve significantly with both quantities. Lines are for visualization and do not imply interpolation.

## 5.3 Comparison with Standard Inference Models & Variational EM

Table 1 contains the estimated marginal log-likelihood on MNIST and CIFAR-10 for standard and iterative inference models, including hierarchical inference models. Iterative inference models were trained by encoding the data and errors for 5 inference iterations. With the *same architecture*, iterative inference models outperform their standard counterparts. See Appendix C.5 for details and discussion. We also compared the inference optimization performance of iterative inference models with variational EM expectation steps using various optimizers. In Figure 6, we observe that the iterative inference model empirically converges substantially faster to *better* estimates, even with only local gradient information. See Appendix C.6 for details and discussion. To summarize, iterative inference models outperform standard inference models in terms of inference capabilities, yet are far more computationally efficient than variational EM.

Table 1: Test set performance on MNIST (in *nats*) and CIFAR-10 (in *bits/input dimension*) for standard and iterative inference models.

|  | $-\log p(\mathbf{x}) \approx$ |
|---|---|
| **MNIST** | |
| *One-Level Model* | |
| Standard (VAE) | $84.14 \pm 0.02$ |
| Iterative | $\mathbf{83.84 \pm 0.05}$ |
| *Hierarchical Model* | |
| Standard (VAE) | $82.63 \pm 0.01$ |
| Iterative | $\mathbf{82.457 \pm 0.001}$ |
| **CIFAR-10** | |
| *One-Level Model* | |
| Standard (VAE) | $5.823 \pm 0.001$ |
| Iterative | $\mathbf{5.71 \pm 0.02}$ |

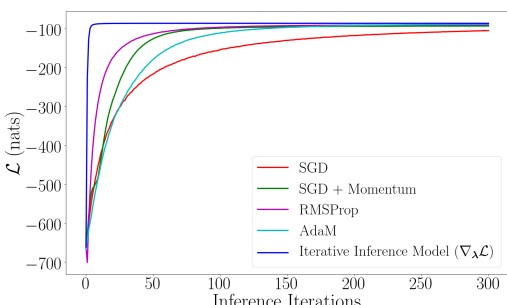

Figure 6: Comparison of inference optimization performance on MNIST test set between iterative inference models and conventional optimization techniques. Iterative inference models empirically converge faster.

## 6 CONCLUSION

We have proposed a new class of inference models, which, by encoding approximate posterior gradients, learn to iteratively refine their inference estimates. These models relate to previous work on VAEs, as well as learning to learn. We have demonstrated that these models can indeed learn to perform approximate posterior optimization, and we have shown the empirical advantages of this approach over current inference techniques on benchmark data sets. Combining iterative inference models with other recent advances in Bayesian deep learning could yield additional insights.

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

# A APPROXIMATE POSTERIOR GRADIENTS FOR LATENT GAUSSIAN MODELS

## A.1 MODEL & OBJECTIVE

Consider a latent variable model, $p_\theta(\mathbf{x}, \mathbf{z}) = p_\theta(\mathbf{x}|\mathbf{z})p_\theta(\mathbf{z})$, where the prior on $\mathbf{z}$ is a factorized Gaussian density, $p_\theta(\mathbf{z}) = \mathcal{N}(\mathbf{z}; \boldsymbol{\mu}_p, \text{diag}\, \boldsymbol{\sigma}_{\mathbf{x}}^2)$, and the conditional likelihood, $p_\theta(\mathbf{x}|\mathbf{z})$, is Bernoulli for binary observations or Gaussian for continuous observations. We introduce an approximate posterior distribution, $q(\mathbf{z}|\mathbf{x})$, which can be any parametric probability density defined over real values. Here, we assume that $q$ also takes the form of a factorized Gaussian density, $q(\mathbf{z}|\mathbf{x}) = \mathcal{N}(\mathbf{z}; \boldsymbol{\mu}_q, \text{diag}\, \boldsymbol{\sigma}_q^2)$. The objective during variational inference is to maximize $\mathcal{L}$ w.r.t. the parameters of $q(\mathbf{z}|\mathbf{x})$, i.e. $\boldsymbol{\mu}_q$ and $\boldsymbol{\sigma}_q^2$:

$$\boldsymbol{\mu}_q^*, \boldsymbol{\sigma}_q^{2*} = \arg\max_{\boldsymbol{\mu}_q, \boldsymbol{\sigma}_q^2} \mathcal{L}. \tag{12}$$

To solve this optimization problem, we will inspect the gradients $\nabla_{\boldsymbol{\mu}_q}\mathcal{L}$ and $\nabla_{\boldsymbol{\sigma}_q^2}\mathcal{L}$, which we now derive. The objective can be written as:

$$\mathcal{L} = \mathbb{E}_{q(\mathbf{z}|\mathbf{x})}\left[\log p_\theta(\mathbf{x}, \mathbf{z}) - \log q(\mathbf{z}|\mathbf{x})\right] \tag{13}$$

$$= \mathbb{E}_{q(\mathbf{z}|\mathbf{x})}\left[\log p_\theta(\mathbf{x}|\mathbf{z}) + \log p_\theta(\mathbf{z}) - \log q(\mathbf{z}|\mathbf{x})\right]. \tag{14}$$

Plugging in $p_\theta(\mathbf{z})$ and $q(\mathbf{z}|\mathbf{x})$:

$$\mathcal{L} = \mathbb{E}_{\mathcal{N}(\mathbf{z}; \boldsymbol{\mu}_q, \text{diag}\, \boldsymbol{\sigma}_q^2)}\left[\log p_\theta(\mathbf{x}|\mathbf{z}) + \log \mathcal{N}(\mathbf{z}; \boldsymbol{\mu}_p, \text{diag}\, \boldsymbol{\sigma}_p^2) - \log \mathcal{N}(\mathbf{z}; \boldsymbol{\mu}_q, \text{diag}\, \boldsymbol{\sigma}_q^2)\right] \tag{15}$$

Since expectation and differentiation are linear operators, we can take the expectation and derivative of each term individually.

## A.2 GRADIENT OF THE LOG-PRIOR

We can write the log-prior as:

$$\log \mathcal{N}(\mathbf{z}; \boldsymbol{\mu}_p, (\text{diag}\, \boldsymbol{\sigma}_p^2) = -\frac{1}{2}\log\left((2\pi)^{n_\mathbf{z}}|\text{diag}\, \boldsymbol{\sigma}_p^2|\right) - \frac{1}{2}(\mathbf{z} - \boldsymbol{\mu}_p)^\intercal(\text{diag}\, \boldsymbol{\sigma}_p^2)^{-1}(\mathbf{z} - \boldsymbol{\mu}_p), \tag{16}$$

where $n_\mathbf{z}$ is the dimensionality of $\mathbf{z}$. We want to evaluate the following terms:

$$\nabla_{\boldsymbol{\mu}_q}\mathbb{E}_{\mathcal{N}(\mathbf{z}; \boldsymbol{\mu}_q, \text{diag}\, \boldsymbol{\sigma}_q^2)}\left[-\frac{1}{2}\log\left((2\pi)^{n_\mathbf{z}}|\text{diag}\, \boldsymbol{\sigma}_p^2|\right) - \frac{1}{2}(\mathbf{z} - \boldsymbol{\mu}_p)^\intercal(\text{diag}\, \boldsymbol{\sigma}_p^2)^{-1}(\mathbf{z} - \boldsymbol{\mu}_p)\right] \tag{17}$$

and

$$\nabla_{\boldsymbol{\sigma}_q^2}\mathbb{E}_{\mathcal{N}(\mathbf{z}; \boldsymbol{\mu}_q, \text{diag}\, \boldsymbol{\sigma}_q^2)}\left[-\frac{1}{2}\log\left((2\pi)^{n_\mathbf{z}}|\text{diag}\, \boldsymbol{\sigma}_p^2|\right) - \frac{1}{2}(\mathbf{z} - \boldsymbol{\mu}_p)^\intercal(\text{diag}\, \boldsymbol{\sigma}_p^2)^{-1}(\mathbf{z} - \boldsymbol{\mu}_p)\right]. \tag{18}$$

To take these derivatives, we will use the reparameterization trick to re-express $\mathbf{z} = \boldsymbol{\mu}_q + \boldsymbol{\sigma}_q \odot \boldsymbol{\epsilon}$, where $\boldsymbol{\epsilon} \sim \mathcal{N}(\mathbf{0}, \mathbf{I})$ is an auxiliary standard Gaussian variable, and $\odot$ denotes the element-wise product. We can now perform the expectations over $\boldsymbol{\epsilon}$, allowing us to bring the gradient operators inside the expectation brackets. The first term in eqs. 17 and 18 does not depend on $\boldsymbol{\mu}_q$ or $\boldsymbol{\sigma}_q^2$, so we can write:

$$\mathbb{E}_{\mathcal{N}(\boldsymbol{\epsilon}; \mathbf{0}, \mathbf{I})}\left[\nabla_{\boldsymbol{\mu}_q}\left(-\frac{1}{2}(\boldsymbol{\mu}_q + \boldsymbol{\sigma}_q \odot \boldsymbol{\epsilon} - \boldsymbol{\mu}_p)^\intercal(\text{diag}\, \boldsymbol{\sigma}_p^2)^{-1}(\boldsymbol{\mu}_q + \boldsymbol{\sigma}_q \odot \boldsymbol{\epsilon} - \boldsymbol{\mu}_p)\right)\right] \tag{19}$$

and

$$\mathbb{E}_{\mathcal{N}(\boldsymbol{\epsilon}; \mathbf{0}, \mathbf{I})}\left[\nabla_{\boldsymbol{\sigma}_q^2}\left(-\frac{1}{2}(\boldsymbol{\mu}_q + \boldsymbol{\sigma}_q \odot \boldsymbol{\epsilon} - \boldsymbol{\mu}_p)^\intercal(\text{diag}\, \boldsymbol{\sigma}_p^2)^{-1}(\boldsymbol{\mu}_q + \boldsymbol{\sigma}_q \odot \boldsymbol{\epsilon} - \boldsymbol{\mu}_p)\right)\right]. \tag{20}$$

To simplify notation, we define the following term:

$$\boldsymbol{\xi} \equiv (\text{diag}\, \boldsymbol{\sigma}_p^2)^{-1/2}(\boldsymbol{\mu}_q + \boldsymbol{\sigma}_q \odot \boldsymbol{\epsilon} - \boldsymbol{\mu}_p), \tag{21}$$

allowing us to rewrite eqs. 19 and 20 as:

$$\mathbb{E}_{\mathcal{N}(\boldsymbol{\epsilon}; \mathbf{0}, \mathbf{I})}\left[\nabla_{\boldsymbol{\mu}_q}\left(-\frac{1}{2}\boldsymbol{\xi}^\intercal\boldsymbol{\xi}\right)\right] = \mathbb{E}_{\mathcal{N}(\boldsymbol{\epsilon}; \mathbf{0}, \mathbf{I})}\left[-\frac{\partial\boldsymbol{\xi}^\intercal}{\partial\boldsymbol{\mu}_q}\boldsymbol{\xi}\right] \tag{22}$$

and

$$\mathbb{E}_{\mathcal{N}(\boldsymbol{\epsilon};\mathbf{0},\mathbf{I})} \left[ \nabla_{\boldsymbol{\sigma}_q^2} \left( -\frac{1}{2} \boldsymbol{\xi}^\mathsf{T} \boldsymbol{\xi} \right) \right] = \mathbb{E}_{\mathcal{N}(\boldsymbol{\epsilon};\mathbf{0},\mathbf{I})} \left[ -\frac{\partial \boldsymbol{\xi}^\mathsf{T}}{\partial \boldsymbol{\sigma}_q^2} \boldsymbol{\xi} \right]. \tag{23}$$

We must now find $\frac{\partial \boldsymbol{\xi}}{\partial \boldsymbol{\mu}_q}$ and $\frac{\partial \boldsymbol{\xi}}{\partial \boldsymbol{\sigma}_q^2}$:

$$\frac{\partial \boldsymbol{\xi}}{\partial \boldsymbol{\mu}_q} = \frac{\partial}{\partial \boldsymbol{\mu}_q} \left( (\operatorname{diag} \boldsymbol{\sigma}_p^2)^{-1/2} (\boldsymbol{\mu}_q + \boldsymbol{\sigma}_q \odot \boldsymbol{\epsilon} - \boldsymbol{\mu}_p) \right) = (\operatorname{diag} \boldsymbol{\sigma}_p^2)^{-1/2} \tag{24}$$

and

$$\frac{\partial \boldsymbol{\xi}}{\partial \boldsymbol{\sigma}_q^2} = \frac{\partial}{\partial \boldsymbol{\sigma}_q^2} \left( (\operatorname{diag} \boldsymbol{\sigma}_p^2)^{-1/2} (\boldsymbol{\mu}_q + \boldsymbol{\sigma}_q \odot \boldsymbol{\epsilon} - \boldsymbol{\mu}_p) \right) = (\operatorname{diag} \boldsymbol{\sigma}_p^2)^{-1/2} \operatorname{diag} \frac{\boldsymbol{\epsilon}}{2\boldsymbol{\sigma}_q}, \tag{25}$$

where division is performed element-wise. Plugging eqs. 24 and 25 back into eqs. 22 and 23, we get:

$$\mathbb{E}_{\mathcal{N}(\boldsymbol{\epsilon};\mathbf{0},\mathbf{I})} \left[ - \left( (\operatorname{diag} \boldsymbol{\sigma}_p^2)^{-1/2} \right)^\mathsf{T} (\operatorname{diag} \boldsymbol{\sigma}_p^2)^{-1/2} (\boldsymbol{\mu}_q + \boldsymbol{\sigma}_q \odot \boldsymbol{\epsilon} - \boldsymbol{\mu}_p) \right] \tag{26}$$

and

$$\mathbb{E}_{\mathcal{N}(\boldsymbol{\epsilon};\mathbf{0},\mathbf{I})} \left[ - \left( \operatorname{diag} \frac{\boldsymbol{\epsilon}}{2\boldsymbol{\sigma}_q} \right)^\mathsf{T} \left( (\operatorname{diag} \boldsymbol{\sigma}_p^2)^{-1/2} \right)^\mathsf{T} (\operatorname{diag} \boldsymbol{\sigma}_p^2)^{-1/2} (\boldsymbol{\mu}_q + \boldsymbol{\sigma}_q \odot \boldsymbol{\epsilon} - \boldsymbol{\mu}_p) \right]. \tag{27}$$

Putting everything together, we can express the gradients as:

$$\nabla_{\boldsymbol{\mu}_q} \mathbb{E}_{\mathcal{N}(\mathbf{z};\boldsymbol{\mu}_q, \operatorname{diag} \boldsymbol{\sigma}_q^2)} \left[ \log \mathcal{N}(\mathbf{z}; \boldsymbol{\mu}_p, \operatorname{diag} \boldsymbol{\sigma}_p^2) \right] = \mathbb{E}_{\mathcal{N}(\boldsymbol{\epsilon};\mathbf{0},\mathbf{I})} \left[ -\frac{\boldsymbol{\mu}_q + \boldsymbol{\sigma}_q \odot \boldsymbol{\epsilon} - \boldsymbol{\mu}_p}{\boldsymbol{\sigma}_p^2} \right], \tag{28}$$

and

$$\nabla_{\boldsymbol{\sigma}_q^2} \mathbb{E}_{\mathcal{N}(\mathbf{z};\boldsymbol{\mu}_q, \operatorname{diag} \boldsymbol{\sigma}_q^2)} \left[ \log \mathcal{N}(\mathbf{z}; \boldsymbol{\mu}_p, \operatorname{diag} \boldsymbol{\sigma}_p^2) \right] =$$
$$\mathbb{E}_{\mathcal{N}(\boldsymbol{\epsilon};\mathbf{0},\mathbf{I})} \left[ - \left( \operatorname{diag} \frac{\boldsymbol{\epsilon}}{2\boldsymbol{\sigma}_q} \right)^\mathsf{T} \frac{\boldsymbol{\mu}_q + \boldsymbol{\sigma}_q \odot \boldsymbol{\epsilon} - \boldsymbol{\mu}_p}{\boldsymbol{\sigma}_p^2} \right]. \tag{29}$$

### A.3 Gradient of the Log-Approximate Posterior

We can write the log-approximate posterior as:

$$\log \mathcal{N}(\mathbf{z}; \boldsymbol{\mu}_q, \operatorname{diag} \boldsymbol{\sigma}_q^2) = -\frac{1}{2} \log \left( (2\pi)^{n_\mathbf{z}} | \operatorname{diag} \boldsymbol{\sigma}_q^2 | \right) - \frac{1}{2} (\mathbf{z} - \boldsymbol{\mu}_q)^\mathsf{T} (\operatorname{diag} \boldsymbol{\sigma}_q^2)^{-1} (\mathbf{z} - \boldsymbol{\mu}_q), \tag{30}$$

where $n_\mathbf{z}$ is the dimensionality of $\mathbf{z}$. Again, we will use the reparameterization trick to re-express the gradients. However, notice what happens when plugging the reparameterized $\mathbf{z} = \boldsymbol{\mu}_q + \boldsymbol{\sigma}_q \odot \boldsymbol{\epsilon}$ into the second term of eq. 30:

$$-\frac{1}{2} (\boldsymbol{\mu}_q + \boldsymbol{\sigma}_q \odot \boldsymbol{\epsilon} - \boldsymbol{\mu}_q)^\mathsf{T} (\operatorname{diag} \boldsymbol{\sigma}_q^2)^{-1} (\boldsymbol{\mu}_q + \boldsymbol{\sigma}_q \odot \boldsymbol{\epsilon} - \boldsymbol{\mu}_q) = -\frac{1}{2} \frac{(\boldsymbol{\sigma}_q \odot \boldsymbol{\epsilon})^\mathsf{T} (\boldsymbol{\sigma}_q \odot \boldsymbol{\epsilon})}{\boldsymbol{\sigma}_q^2} = -\frac{1}{2} \boldsymbol{\epsilon}^\mathsf{T} \boldsymbol{\epsilon}. \tag{31}$$

This term does not depend on $\boldsymbol{\mu}_q$ or $\boldsymbol{\sigma}_q^2$. Also notice that the first term in eq. 30 depends only on $\boldsymbol{\sigma}_q^2$. Therefore, the gradient of the entire term w.r.t. $\boldsymbol{\mu}_q$ is zero:

$$\nabla_{\boldsymbol{\mu}_q} \mathbb{E}_{\mathcal{N}(\mathbf{z};\boldsymbol{\mu}_q, \operatorname{diag} \boldsymbol{\sigma}_q^2)} \left[ \log \mathcal{N}(\mathbf{z}; \boldsymbol{\mu}_q, \operatorname{diag} \boldsymbol{\sigma}_q^2) \right] = \mathbf{0}. \tag{32}$$

The gradient w.r.t. $\boldsymbol{\sigma}_q^2$ is

$$\nabla_{\boldsymbol{\sigma}_q^2} \left( -\frac{1}{2} \log \left( (2\pi)^{n_\mathbf{z}} | \operatorname{diag} \boldsymbol{\sigma}_q^2 | \right) \right) = -\frac{1}{2} \nabla_{\boldsymbol{\sigma}_q^2} \left( \log | \operatorname{diag} \boldsymbol{\sigma}_q^2 | \right) = -\frac{1}{2} \nabla_{\boldsymbol{\sigma}_q^2} \sum_j \log \sigma_{q,j}^2 = -\frac{1}{2\boldsymbol{\sigma}_q^2}. \tag{33}$$

Note that the expectation has been dropped, as the term does not depend on the value of the sampled $\mathbf{z}$. Thus, the gradient of the entire term w.r.t. $\boldsymbol{\sigma}_q^2$ is:

$$\nabla_{\boldsymbol{\sigma}_q^2} \mathbb{E}_{\mathcal{N}(\mathbf{z};\boldsymbol{\mu}_q, \operatorname{diag} \boldsymbol{\sigma}_q^2)} \left[ \log \mathcal{N}(\mathbf{z}; \boldsymbol{\mu}_q, \operatorname{diag} \boldsymbol{\sigma}_q^2) \right] = -\frac{1}{2\boldsymbol{\sigma}_q^2}. \tag{34}$$

A.4    GRADIENT OF THE LOG-CONDITIONAL LIKELIHOOD

The form of the conditional likelihood will depend on the data, e.g. binary, discrete, continuous, etc. Here, we derive the gradient for Bernoulli (binary) and Gaussian (continuous) conditional likelihoods.

**Bernoulli Output Distribution**    The log of a Bernoulli output distribution takes the form:

$$\log \mathcal{B}(\mathbf{x}; \boldsymbol{\mu}_\mathbf{x}) = (\log \boldsymbol{\mu}_\mathbf{x})^\mathsf{T} \mathbf{x} + (\log(\mathbf{1} - \boldsymbol{\mu}_\mathbf{x}))^\mathsf{T} (\mathbf{1} - \mathbf{x}), \tag{35}$$

where $\boldsymbol{\mu}_\mathbf{x} = \boldsymbol{\mu}_\mathbf{x}(\mathbf{z}, \theta)$ is the mean of the output distribution. We drop the explicit dependence on $\mathbf{z}$ and $\theta$ to simplify notation. We want to compute the gradients

$$\nabla_{\boldsymbol{\mu}_q} \mathbb{E}_{\mathcal{N}(\mathbf{z}; \boldsymbol{\mu}_q, \mathrm{diag}\,\boldsymbol{\sigma}_q^2)} \left[ (\log \boldsymbol{\mu}_\mathbf{x})^\mathsf{T} \mathbf{x} + (\log(\mathbf{1} - \boldsymbol{\mu}_\mathbf{x}))^\mathsf{T} (\mathbf{1} - \mathbf{x}) \right] \tag{36}$$

and

$$\nabla_{\boldsymbol{\sigma}_q^2} \mathbb{E}_{\mathcal{N}(\mathbf{z}; \boldsymbol{\mu}_q, \mathrm{diag}\,\boldsymbol{\sigma}_q^2)} \left[ (\log \boldsymbol{\mu}_\mathbf{x})^\mathsf{T} \mathbf{x} + (\log(\mathbf{1} - \boldsymbol{\mu}_\mathbf{x}))^\mathsf{T} (\mathbf{1} - \mathbf{x}) \right]. \tag{37}$$

Again, we use the reparameterization trick to re-express the expectations, allowing us to bring the gradient operators inside the brackets. Using $\mathbf{z} = \boldsymbol{\mu}_q + \boldsymbol{\sigma}_q \odot \boldsymbol{\epsilon}$, eqs. 36 and 37 become:

$$\mathbb{E}_{\mathcal{N}(\boldsymbol{\epsilon}; \mathbf{0}, \mathbf{I})} \left[ \nabla_{\boldsymbol{\mu}_q} \left( (\log \boldsymbol{\mu}_\mathbf{x})^\mathsf{T} \mathbf{x} + (\log(\mathbf{1} - \boldsymbol{\mu}_\mathbf{x}))^\mathsf{T} (\mathbf{1} - \mathbf{x}) \right) \right] \tag{38}$$

and

$$\mathbb{E}_{\mathcal{N}(\boldsymbol{\epsilon}; \mathbf{0}, \mathbf{I})} \left[ \nabla_{\boldsymbol{\sigma}_q^2} \left( (\log \boldsymbol{\mu}_\mathbf{x})^\mathsf{T} \mathbf{x} + (\log(\mathbf{1} - \boldsymbol{\mu}_\mathbf{x}))^\mathsf{T} (\mathbf{1} - \mathbf{x}) \right) \right], \tag{39}$$

where $\boldsymbol{\mu}_\mathbf{x}$ is re-expressed as function of $\boldsymbol{\mu}_q, \boldsymbol{\sigma}_q^2, \boldsymbol{\epsilon}$, and $\theta$. Distributing the gradient operators yields:

$$\mathbb{E}_{\mathcal{N}(\boldsymbol{\epsilon}; \mathbf{0}, \mathbf{I})} \left[ \frac{\partial (\log \boldsymbol{\mu}_\mathbf{x})^\mathsf{T}}{\partial \boldsymbol{\mu}_q} \mathbf{x} + \frac{\partial (\log(\mathbf{1} - \boldsymbol{\mu}_\mathbf{x}))^\mathsf{T}}{\partial \boldsymbol{\mu}_q} (\mathbf{1} - \mathbf{x}) \right] \tag{40}$$

and

$$\mathbb{E}_{\mathcal{N}(\boldsymbol{\epsilon}; \mathbf{0}, \mathbf{I})} \left[ \frac{\partial (\log \boldsymbol{\mu}_\mathbf{x})^\mathsf{T}}{\partial \boldsymbol{\sigma}_q^2} \mathbf{x} + \frac{\partial (\log(\mathbf{1} - \boldsymbol{\mu}_\mathbf{x}))^\mathsf{T}}{\partial \boldsymbol{\sigma}_q^2} (\mathbf{1} - \mathbf{x}) \right]. \tag{41}$$

Taking the partial derivatives and combining terms gives:

$$\mathbb{E}_{\mathcal{N}(\boldsymbol{\epsilon}; \mathbf{0}, \mathbf{I})} \left[ \frac{\partial \boldsymbol{\mu}_\mathbf{x}}{\partial \boldsymbol{\mu}_q}^\mathsf{T} \frac{\mathbf{x}}{\boldsymbol{\mu}_\mathbf{x}} - \frac{\partial \boldsymbol{\mu}_\mathbf{x}}{\partial \boldsymbol{\mu}_q}^\mathsf{T} \frac{\mathbf{1} - \mathbf{x}}{\mathbf{1} - \boldsymbol{\mu}_\mathbf{x}} \right] = \mathbb{E}_{\mathcal{N}(\boldsymbol{\epsilon}; \mathbf{0}, \mathbf{I})} \left[ \frac{\partial \boldsymbol{\mu}_\mathbf{x}}{\partial \boldsymbol{\mu}_q}^\mathsf{T} \frac{\mathbf{x} - \boldsymbol{\mu}_\mathbf{x}}{\boldsymbol{\mu}_\mathbf{x} \odot (\mathbf{1} - \boldsymbol{\mu}_\mathbf{x})} \right] \tag{42}$$

and

$$\mathbb{E}_{\mathcal{N}(\boldsymbol{\epsilon}; \mathbf{0}, \mathbf{I})} \left[ \frac{\partial \boldsymbol{\mu}_\mathbf{x}}{\partial \boldsymbol{\sigma}_q^2}^\mathsf{T} \frac{\mathbf{x}}{\boldsymbol{\mu}_\mathbf{x}} - \frac{\partial \boldsymbol{\mu}_\mathbf{x}}{\partial \boldsymbol{\sigma}_q^2}^\mathsf{T} \frac{\mathbf{1} - \mathbf{x}}{\mathbf{1} - \boldsymbol{\mu}_\mathbf{x}} \right] = \mathbb{E}_{\mathcal{N}(\boldsymbol{\epsilon}; \mathbf{0}, \mathbf{I})} \left[ \frac{\partial \boldsymbol{\mu}_\mathbf{x}}{\partial \boldsymbol{\sigma}_q^2}^\mathsf{T} \frac{\mathbf{x} - \boldsymbol{\mu}_\mathbf{x}}{\boldsymbol{\mu}_\mathbf{x} \odot (\mathbf{1} - \boldsymbol{\mu}_\mathbf{x})} \right]. \tag{43}$$

**Gaussian Output Density**    The log of a Gaussian output density takes the form:

$$\log \mathcal{N}(\mathbf{x}; \boldsymbol{\mu}_\mathbf{x}, \mathrm{diag}\,\boldsymbol{\sigma}_\mathbf{x}^2) = -\frac{1}{2} \log \left( (2\pi)^{n_\mathbf{x}} |\mathrm{diag}\,\boldsymbol{\sigma}_\mathbf{x}^2| \right) - \frac{1}{2} (\mathbf{x} - \boldsymbol{\mu}_\mathbf{x})^\mathsf{T} (\mathrm{diag}\,\boldsymbol{\sigma}_\mathbf{x}^2)^{-1} (\mathbf{x} - \boldsymbol{\mu}_\mathbf{x}), \tag{44}$$

where $\boldsymbol{\mu}_\mathbf{x} = \boldsymbol{\mu}_\mathbf{x}(\mathbf{z}, \theta)$ is the mean of the output distribution and $\boldsymbol{\sigma}_\mathbf{x}^2 = \boldsymbol{\sigma}_\mathbf{x}^2(\theta)$ is the variance. We assume $\boldsymbol{\sigma}_\mathbf{x}^2$ is not a function of $\mathbf{z}$ to simplify the derivation, however, using $\boldsymbol{\sigma}_\mathbf{x}^2 = \boldsymbol{\sigma}_\mathbf{x}^2(\mathbf{z}, \theta)$ is possible and would simply result in additional gradient terms in $\nabla_{\boldsymbol{\mu}_q} \mathcal{L}$ and $\nabla_{\boldsymbol{\sigma}_q^2} \mathcal{L}$. We want to compute the gradients

$$\nabla_{\boldsymbol{\mu}_q} \mathbb{E}_{\mathcal{N}(\mathbf{z}; \boldsymbol{\mu}_q, \mathrm{diag}\,\boldsymbol{\sigma}_q^2)} \left[ -\frac{1}{2} \log \left( (2\pi)^{n_\mathbf{x}} |\mathrm{diag}\,\boldsymbol{\sigma}_\mathbf{x}^2| \right) - \frac{1}{2} (\mathbf{x} - \boldsymbol{\mu}_\mathbf{x})^\mathsf{T} (\mathrm{diag}\,\boldsymbol{\sigma}_\mathbf{x}^2)^{-1} (\mathbf{x} - \boldsymbol{\mu}_\mathbf{x}) \right] \tag{45}$$

and

$$\nabla_{\boldsymbol{\sigma}_q^2} \mathbb{E}_{\mathcal{N}(\mathbf{z}; \boldsymbol{\mu}_q, \mathrm{diag}\,\boldsymbol{\sigma}_q^2)} \left[ -\frac{1}{2} \log \left( (2\pi)^{n_\mathbf{x}} |\mathrm{diag}\,\boldsymbol{\sigma}_\mathbf{x}^2| \right) - \frac{1}{2} (\mathbf{x} - \boldsymbol{\mu}_\mathbf{x})^\mathsf{T} (\mathrm{diag}\,\boldsymbol{\sigma}_\mathbf{x}^2)^{-1} (\mathbf{x} - \boldsymbol{\mu}_\mathbf{x}) \right]. \tag{46}$$

The first term in eqs. 45 and 46 is zero, since $\boldsymbol{\sigma}_\mathbf{x}^2$ does not depend on $\boldsymbol{\mu}_q$ or $\boldsymbol{\sigma}_q^2$. To take the gradients, we will again use the reparameterization trick to re-express $\mathbf{z} = \boldsymbol{\mu}_q + \boldsymbol{\sigma}_q \odot \boldsymbol{\epsilon}$. We now implicitly express $\boldsymbol{\mu}_\mathbf{x}$ as $\boldsymbol{\mu}_\mathbf{x}(\boldsymbol{\mu}_q, \boldsymbol{\sigma}_q^2, \theta)$. We can then write:

$$\mathbb{E}_{\mathcal{N}(\boldsymbol{\epsilon}; \mathbf{0}, \mathbf{I})} \left[ \nabla_{\boldsymbol{\mu}_q} \left( -\frac{1}{2} (\mathbf{x} - \boldsymbol{\mu}_\mathbf{x})^\mathsf{T} (\mathrm{diag}\,\boldsymbol{\sigma}_\mathbf{x}^2)^{-1} (\mathbf{x} - \boldsymbol{\mu}_\mathbf{x}) \right) \right] \tag{47}$$

and

$$\mathbb{E}_{\mathcal{N}(\boldsymbol{\epsilon};\mathbf{0},\mathbf{I})}\left[\nabla_{\sigma_q^2}\left(-\frac{1}{2}(\mathbf{x}-\boldsymbol{\mu_x})^\intercal(\operatorname{diag}\boldsymbol{\sigma_x^2})^{-1}(\mathbf{x}-\boldsymbol{\mu_x})\right)\right]. \tag{48}$$

To simplify notation, we define the following term:

$$\boldsymbol{\xi} \equiv (\operatorname{diag}\boldsymbol{\sigma_x^2})^{-1/2}(\mathbf{x}-\boldsymbol{\mu_x}), \tag{49}$$

allowing us to rewrite eqs. 47 and 48 as

$$\mathbb{E}_{\mathcal{N}(\boldsymbol{\epsilon};\mathbf{0},\mathbf{I})}\left[\nabla_{\boldsymbol{\mu}_q}\left(-\frac{1}{2}\boldsymbol{\xi}^\intercal\boldsymbol{\xi}\right)\right] = \mathbb{E}_{\mathcal{N}(\boldsymbol{\epsilon};\mathbf{0},\mathbf{I})}\left[-\frac{\partial\boldsymbol{\xi}^\intercal}{\partial\boldsymbol{\mu}_q}\boldsymbol{\xi}\right] \tag{50}$$

and

$$\mathbb{E}_{\mathcal{N}(\boldsymbol{\epsilon};\mathbf{0},\mathbf{I})}\left[\nabla_{\sigma_q^2}\left(-\frac{1}{2}\boldsymbol{\xi}^\intercal\boldsymbol{\xi}\right)\right] = \mathbb{E}_{\mathcal{N}(\boldsymbol{\epsilon};\mathbf{0},\mathbf{I})}\left[-\frac{\partial\boldsymbol{\xi}^\intercal}{\partial\boldsymbol{\sigma}_q^2}\boldsymbol{\xi}\right]. \tag{51}$$

We must now find $\frac{\partial\boldsymbol{\xi}}{\partial\boldsymbol{\mu}_q}$ and $\frac{\partial\boldsymbol{\xi}}{\partial\boldsymbol{\sigma}_q^2}$:

$$\frac{\partial\boldsymbol{\xi}}{\partial\boldsymbol{\mu}_q} = \frac{\partial}{\partial\boldsymbol{\mu}_q}\left((\operatorname{diag}\boldsymbol{\sigma_x^2})^{-1/2}(\mathbf{x}-\boldsymbol{\mu_x})\right) = -(\operatorname{diag}\boldsymbol{\sigma_x^2})^{-1/2}\frac{\partial\boldsymbol{\mu_x}}{\partial\boldsymbol{\mu}_q} \tag{52}$$

and

$$\frac{\partial\boldsymbol{\xi}}{\partial\boldsymbol{\sigma}_q^2} = \frac{\partial}{\partial\boldsymbol{\sigma}_q^2}\left((\operatorname{diag}\boldsymbol{\sigma_x^2})^{-1/2}(\mathbf{x}-\boldsymbol{\mu_x})\right) = -(\operatorname{diag}\boldsymbol{\sigma_x^2})^{-1/2}\frac{\partial\boldsymbol{\mu_x}}{\partial\boldsymbol{\sigma}_q^2}. \tag{53}$$

Plugging these expressions back into eqs. 50 and 51 gives

$$\mathbb{E}_{\mathcal{N}(\boldsymbol{\epsilon};\mathbf{0},\mathbf{I})}\left[\frac{\partial\boldsymbol{\mu_x}}{\partial\boldsymbol{\mu}_q}^\intercal((\operatorname{diag}\boldsymbol{\sigma_x^2})^{-1/2})^\intercal(\operatorname{diag}\boldsymbol{\sigma_x^2})^{-1/2}(\mathbf{x}-\boldsymbol{\mu_x})\right] = \mathbb{E}_{\mathcal{N}(\boldsymbol{\epsilon};\mathbf{0},\mathbf{I})}\left[\frac{\partial\boldsymbol{\mu_x}}{\partial\boldsymbol{\mu}_q}^\intercal\frac{\mathbf{x}-\boldsymbol{\mu_x}}{\boldsymbol{\sigma_x^2}}\right] \tag{54}$$

and

$$\mathbb{E}_{\mathcal{N}(\boldsymbol{\epsilon};\mathbf{0},\mathbf{I})}\left[\frac{\partial\boldsymbol{\mu_x}}{\partial\boldsymbol{\sigma}_q^2}^\intercal((\operatorname{diag}\boldsymbol{\sigma_x^2})^{-1/2})^\intercal(\operatorname{diag}\boldsymbol{\sigma_x^2})^{-1/2}(\mathbf{x}-\boldsymbol{\mu_x})\right] = \mathbb{E}_{\mathcal{N}(\boldsymbol{\epsilon};\mathbf{0},\mathbf{I})}\left[\frac{\partial\boldsymbol{\mu_x}}{\partial\boldsymbol{\sigma}_q^2}^\intercal\frac{\mathbf{x}-\boldsymbol{\mu_x}}{\boldsymbol{\sigma_x^2}}\right]. \tag{55}$$

Despite having different distribution forms, Bernoulli and Gaussian output distributions result in approximate posterior gradients of a similar form: the Jacobian of the output model multiplied by a weighted error term.

## A.5 SUMMARY

Putting the gradient terms from $\log p_\theta(\mathbf{x}|\mathbf{z})$, $\log p_\theta(\mathbf{z})$, and $\log q(\mathbf{z}|\mathbf{x})$ together, we arrive at

**Bernoulli Output Distribution**:

$$\nabla_{\boldsymbol{\mu}_q}\mathcal{L} = \mathbb{E}_{\mathcal{N}(\boldsymbol{\epsilon};\mathbf{0},\mathbf{I})}\left[\frac{\partial\boldsymbol{\mu_x}}{\partial\boldsymbol{\mu}_q}^\intercal\frac{\mathbf{x}-\boldsymbol{\mu_x}}{\boldsymbol{\mu_x}\odot(\mathbf{1}-\boldsymbol{\mu_x})} - \frac{\boldsymbol{\mu}_q+\boldsymbol{\sigma}_q\odot\boldsymbol{\epsilon}-\boldsymbol{\mu}_p}{\boldsymbol{\sigma}_p^2}\right] \tag{56}$$

$$\nabla_{\sigma_q^2}\mathcal{L} = \mathbb{E}_{\mathcal{N}(\boldsymbol{\epsilon};\mathbf{0},\mathbf{I})}\left[\frac{\partial\boldsymbol{\mu_x}}{\partial\boldsymbol{\sigma}_q^2}^\intercal\frac{\mathbf{x}-\boldsymbol{\mu_x}}{\boldsymbol{\mu_x}\odot(\mathbf{1}-\boldsymbol{\mu_x})} - \left(\operatorname{diag}\frac{\boldsymbol{\epsilon}}{2\boldsymbol{\sigma}_q}\right)^\intercal\frac{\boldsymbol{\mu}_q+\boldsymbol{\sigma}_q\odot\boldsymbol{\epsilon}-\boldsymbol{\mu}_p}{\boldsymbol{\sigma}_p^2}\right] - \frac{1}{2\boldsymbol{\sigma}_q^2} \tag{57}$$

**Gaussian Output Distribution**:

$$\nabla_{\boldsymbol{\mu}_q}\mathcal{L} = \mathbb{E}_{\mathcal{N}(\boldsymbol{\epsilon};\mathbf{0},\mathbf{I})}\left[\frac{\partial\boldsymbol{\mu_x}}{\partial\boldsymbol{\mu}_q}^\intercal\frac{\mathbf{x}-\boldsymbol{\mu_x}}{\boldsymbol{\sigma_x^2}} - \frac{\boldsymbol{\mu}_q+\boldsymbol{\sigma}_q\odot\boldsymbol{\epsilon}-\boldsymbol{\mu}_p}{\boldsymbol{\sigma}_p^2}\right] \tag{58}$$

$$\nabla_{\sigma_q^2}\mathcal{L} = \mathbb{E}_{\mathcal{N}(\boldsymbol{\epsilon};\mathbf{0},\mathbf{I})}\left[\frac{\partial\boldsymbol{\mu_x}}{\partial\boldsymbol{\sigma}_q^2}^\intercal\frac{\mathbf{x}-\boldsymbol{\mu_x}}{\boldsymbol{\sigma_x^2}} - \left(\operatorname{diag}\frac{\boldsymbol{\epsilon}}{2\boldsymbol{\sigma}_q}\right)^\intercal\frac{\boldsymbol{\mu}_q+\boldsymbol{\sigma}_q\odot\boldsymbol{\epsilon}-\boldsymbol{\mu}_p}{\boldsymbol{\sigma}_p^2}\right] - \frac{1}{2\boldsymbol{\sigma}_q^2} \tag{59}$$

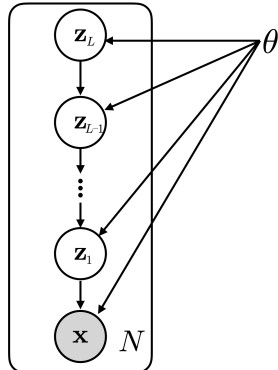

Figure 7: Plate notation for a hierarchical latent variable model consisting of $L$ levels of latent variables. Variables at higher levels provide empirical priors on variables at lower levels. With data-dependent priors, the model has more flexibility in representing the intricacies of each data example.

## A.6 Approximate Posterior Gradients in Hierarchical Models

Hierarchical latent variable models factorize the latent variables over multiple levels, $\mathbf{z} = \{\mathbf{z}_1, \mathbf{z}_2, \ldots, \mathbf{z}_L\}$. Latent variables at higher levels provide *empirical priors* on latent variables at lower levels. For an intermediate latent level, we use the notation $q(\mathbf{z}_\ell|\cdot) = \mathcal{N}(\mathbf{z}_\ell; \boldsymbol{\mu}_{\ell,q}, \text{diag}\,\boldsymbol{\sigma}^2_{\ell,q})$ and $p(\mathbf{z}_\ell|\mathbf{z}_{\ell+1}) = \mathcal{N}(\mathbf{z}_\ell; \boldsymbol{\mu}_{\ell,p}, \text{diag}\,\boldsymbol{\sigma}^2_{\ell,p})$ to denote the approximate posterior and prior respectively. If we assume a strict hierarchy, i.e. $\mathbf{z}_L \rightarrow \mathbf{z}_{L-1} \rightarrow \cdots \rightarrow \mathbf{z}_1 \rightarrow \mathbf{x}$, then the approximate posterior gradients at an intermediate level $\ell$ are:

$$\nabla_{\boldsymbol{\mu}_{q,\ell}}\mathcal{L} = \mathbb{E}_{\mathcal{N}(\boldsymbol{\epsilon};\mathbf{0},\mathbf{I})}\left[\frac{\partial \boldsymbol{\mu}_{\ell-1,p}}{\partial \boldsymbol{\mu}_{\ell,q}}^\mathsf{T} \frac{\boldsymbol{\mu}_{\ell-1,q} + \boldsymbol{\sigma}_{\ell-1,q} \odot \boldsymbol{\epsilon}_{\ell-1} - \boldsymbol{\mu}_{\ell-1,p}}{\boldsymbol{\sigma}^2_{\ell-1,p}}\right.$$
$$\left. - \frac{\boldsymbol{\mu}_{\ell,q} + \boldsymbol{\sigma}_{\ell,q} \odot \boldsymbol{\epsilon}_\ell - \boldsymbol{\mu}_{\ell,p}}{\boldsymbol{\sigma}^2_{\ell,p}}\right], \quad (60)$$

$$\nabla_{\boldsymbol{\sigma}^2_q}\mathcal{L} = \mathbb{E}_{\mathcal{N}(\boldsymbol{\epsilon};\mathbf{0},\mathbf{I})}\left[\frac{\partial \boldsymbol{\mu}_{\ell-1,p}}{\partial \boldsymbol{\sigma}^2_{\ell,q}}^\mathsf{T} \frac{\boldsymbol{\mu}_{\ell-1,q} + \boldsymbol{\sigma}_{\ell-1,q} \odot \boldsymbol{\epsilon}_{\ell-1} - \boldsymbol{\mu}_{\ell-1,p}}{\boldsymbol{\sigma}^2_{\ell-1,p}}\right.$$
$$\left. - \left(\text{diag}\,\frac{\boldsymbol{\epsilon}_\ell}{2\boldsymbol{\sigma}_{\ell,q}}\right)^\mathsf{T} \frac{\boldsymbol{\mu}_{\ell,q} + \boldsymbol{\sigma}_{\ell,q} \odot \boldsymbol{\epsilon}_\ell - \boldsymbol{\mu}_{\ell,p}}{\boldsymbol{\sigma}^2_{\ell,p}}\right] - \frac{1}{2\boldsymbol{\sigma}^2_{\ell,q}}. \quad (61)$$

Notice that these gradients take a similar form to those of a one-level latent variable model. The first terms inside each expectation can be interpreted as a "bottom-up" gradient coming from reconstruction errors at the level below. The second terms inside the expectations can be interpreted as "top-down" errors coming from priors generated by the level above. The last term in the variance gradient expresses a form of regularization. Standard hierarchical inference models only contain bottom-up information, and therefore have no way of estimating the second term in each of these gradients.

## B Implementing Iterative Inference Models

Equation 5 provides a general form for an iterative inference model. Here, we provide specific implementation details for these models. Code for reproducing the experiments will be released online.

## B.1 INPUT FORM

As mentioned in Andrychowicz et al. (2016), gradients can be on vastly different scales, which is undesirable for training neural networks. To handle this issue, we adopt the technique they proposed: replacing $\nabla_{\boldsymbol{\lambda}} \mathcal{L}$ with the concatenation of $[\alpha \log(|\nabla_{\boldsymbol{\lambda}} \mathcal{L}| + \epsilon), \text{sign}(\nabla_{\boldsymbol{\lambda}} \mathcal{L})]$, where $\alpha$ is a scaling constant and $\epsilon$ is a small constant for numerical stability. This is performed for both parameters in $\boldsymbol{\lambda} = \{\boldsymbol{\mu}_q, \log \boldsymbol{\sigma}_q^2\}$. When encoding the errors, we instead input the concatenation of $[\boldsymbol{\varepsilon}_{\mathbf{x}}, \boldsymbol{\varepsilon}_{\mathbf{z}}]$ (see section 4.1 for definitions of these terms). As we use global variances on the output and prior densities, we drop $\boldsymbol{\sigma}_{\mathbf{x}}^2$ and $\boldsymbol{\sigma}_p^2$ from these expressions because they are constant across all examples. We also found it beneficial to encode the current estimates of $\boldsymbol{\mu}_q$ and $\log \boldsymbol{\sigma}_q^2$. We end by again noting that encoding gradients or errors over successive iterations can be difficult, as the distributions of these inputs change quickly during both learning and inference. Work remains to be done in developing iterative encoding architectures that handle this aspect more thoroughly, perhaps through some form of input normalization or saturation.

## B.2 OUTPUT FORM

For the output form of these models, we use a gated updating scheme, sometimes referred to as a "highway" connection (Srivastava et al. (2015)). Specifically, approximate posterior parameters are updated according to

$$\boldsymbol{\lambda}_{t+1} = \mathbf{g}_t \odot \boldsymbol{\lambda}_t + (\mathbf{1} - \mathbf{g}_t) \odot f_t(\nabla_{\boldsymbol{\lambda}} \mathcal{L}, \boldsymbol{\lambda}_t; \phi), \tag{62}$$

where $\odot$ represents element-wise multiplication and $\mathbf{g}_t = g_t(\nabla_{\boldsymbol{\lambda}} \mathcal{L}, \boldsymbol{\lambda}_t; \phi) \in [0, 1]$ is the gating function for $\boldsymbol{\lambda}$ at time $t$, which we combine with the iterative inference model $f_t$. We found that this yielded improved performance and stability over the residual updating scheme used in Andrychowicz et al. (2016). In our experiments with latent Gaussian models, we found that means tend to receive updates over many iterations, whereas variances (or log variances) tend to receive far fewer updates, often just a single large update. Further work could perhaps be done in developing schemes that update these two sets of parameters differently.

## B.3 MODEL FORM

We parameterize iterative inference models as neural networks. Although Andrychowicz et al. (2016) exclusively use recurrent neural networks, we note that optimization models can also be instantiated with feed-forward networks. Note that even with a feed-forward network, because the entire model is run over multiple iterations, the model is technically a recurrent network, though quite different from the standard RNN formulation. RNN iterative inference models, through hidden or memory states, are able to account for non-local curvature information, analogous to momentum or other moment terms in conventional optimization techniques. Feed-forward networks are unable to capture and utilize this information, but purely local curvature information is still sufficient to update the output estimate, e.g. vanilla stochastic gradient descent. Andrychowicz et al. (2016) propagate optimizer parameter gradients ($\nabla_\phi \mathcal{L}$) from the optimizee's loss at each optimization step, giving each step equal weight. We take the same approach; we found it aids in training recurrent iterative inference models and is essential for training feed-forward iterative inference models. With a recurrent model, $\nabla_\phi \mathcal{L}$ is calculated using stochastic backpropagation through time. With a feed-forward model, we accumulate $\nabla_\phi \mathcal{L}$ at each step using stochastic backpropagation, then average over the total number of steps. The advantage of using a feed-forward iterative inference model is that it maintains a constant memory footprint, as we do not need to keep track of gradients across iterations. However, as mentioned above, this limits the iterative inference model to only local optimization information.

## B.4 TRAINING

Overall, we found iterative inference models were not difficult to train. Almost immediately, these models started learning to improve their estimates. As noted by Andrychowicz et al. (2016), some care must be taken to ensure that the input gradients stay within a reasonable range. We found their $\log$ transformation trick to work well in accomplishing this. We also observed that the level of stochasticity in the gradients played a larger role in inference performance for iterative inference

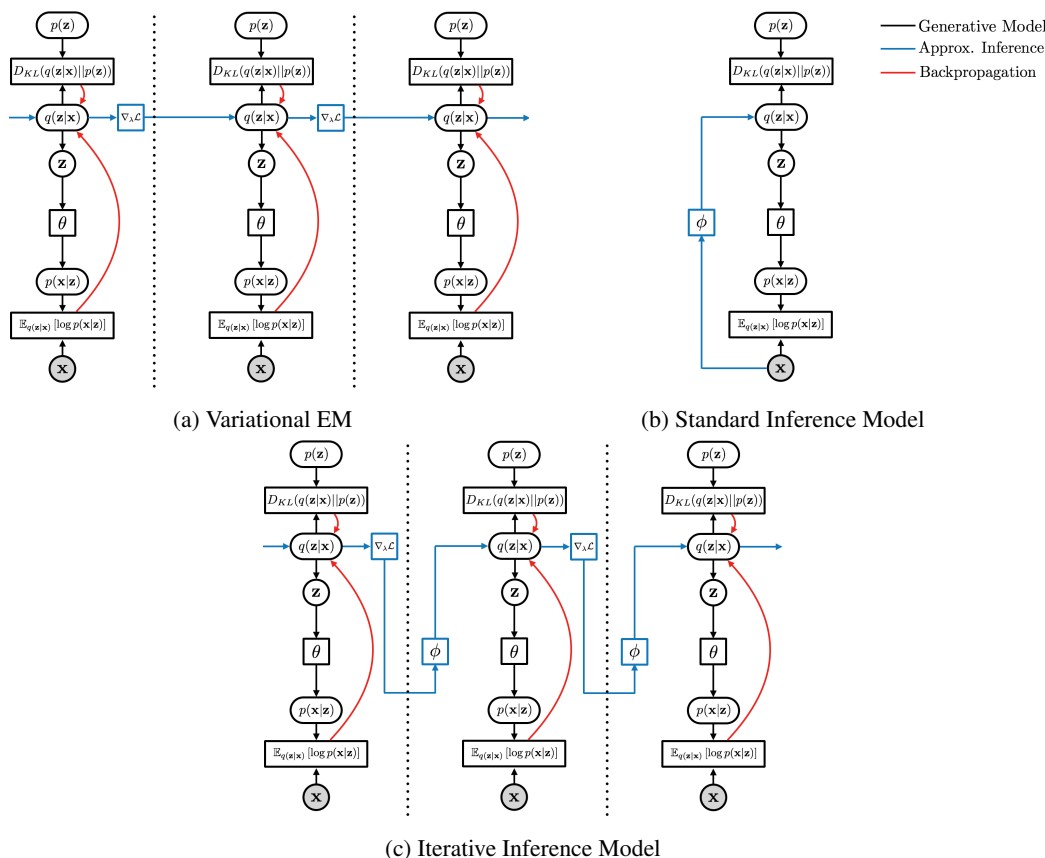

Figure 8: Computational graphs for variational inference with (a) Variational EM, (b) Standard Inference Models, and (c) Iterative Inference Models.

models. For instance, in the Gaussian case, we noticed a sizable difference in performance between approximating the KL-divergence and evaluating it analytically. This difference was much less noticeable for standard inference models.

## C EXPERIMENT DETAILS

In all experiments, inference model and generative model parameters were learned jointly using the AdaM optimizer (Kingma & Ba (2014)). The learning rate was set to 0.0002 for both sets of parameters and all other optimizer parameters were set to their default values. Learning rates were decayed exponentially by a factor of 0.999 at every epoch. All models utilized exponential linear unit (ELU) activation functions (Clevert et al. (2015)), although we found other non-linearities to work as well. Unless otherwise stated, all inference models were symmetric to their corresponding generative models, with the addition of "highway" connections (Srivastava et al. (2015)) between hidden layers. Though not essential, we found that these connections improved stability and performance. Iterative inference models for all experiments were implemented as feed-forward networks to make comparison with standard inference models easier. See appendix B for further details.

### C.1 TWO-DIMENSIONAL LATENT GAUSSIAN MODELS

To visualize the optimization surface and trajectories of latent Gaussian models, we trained models with 2 latent dimensions and a point estimate approximate posterior. That is, $q(\mathbf{z}|\mathbf{x}) = \delta(\mathbf{z} = \boldsymbol{\mu}_q)$ is a Dirac delta function at the point $\boldsymbol{\mu}_q = (\mu_1, \mu_2)$. We used a 2D point estimate approximate posterior instead of a 1D Gaussian density because it results in more variety in the optimization surface, making it easier to visualize the optimization. We trained these models on binarized MNIST

due to the data set's relatively low complexity, meaning that 2 latent dimensions can reasonably capture the relevant information specific to a data example. The generative models consisted of a neural network with 2 hidden layers, each with 512 units. The output of the generative model was the mean of a Bernoulli distribution, and $\log p_\theta(\mathbf{x}|\mathbf{z})$ was evaluated using binary cross-entropy. KL-divergences were estimated using 1 sample of $\mathbf{z} \sim q(\mathbf{z}|\mathbf{x})$. The optimization surface of each model was evaluated on a grid with range [-5, 5] in increments of 0.05 for each latent variable. To approximate the MAP estimate, we up-sampled the optimization surface using a cubic interpolation scheme. Figure 1 visualizes the ELBO optimization surface after training for 80 epochs. Figure 3 visualizes the ELBO optimization surface after training (by encoding $\mathbf{x}$, $\varepsilon_\mathbf{x}$, and $\varepsilon_\mathbf{z}$) for 50 epochs.

## C.2 RECONSTRUCTIONS OVER INFERENCE ITERATIONS

For the qualitative results shown in figure 4, we trained iterative inference models on MNIST, Omniglot, and Street View House Numbers by encoding approximate posterior gradients ($\nabla_\lambda \mathcal{L}$) for 16 inference iterations. For CIFAR-10, we had difficulty in obtaining sharp reconstructions in a reasonable number of inference iterations, so we trained an iterative inference model by encoding errors for 10 inference iterations. For binarized MNIST and Omniglot, we used a generative model architecture with 2 hidden layers, each with 512 units, a latent space of size 64, and a symmetric iterative inference model, with the addition of highway connections at each layer. For Street View House Numbers and CIFAR-10, we used 3 hidden layers in the iterative inference and 1 in the generative model, with 2048 units at each hidden layer and a latent space of size 1024.

## C.3 ADDITIONAL LATENT SAMPLES

We used the same architecture of 2 hidden layers, each with 512 units, for the output model and inference models. The latent variables consisted of 64 dimensions. Each model was trained by drawing the corresponding number of samples from the approximate posterior distribution using the reparameterization trick, yielding lower variance ELBO estimates and gradients. Iterative inference models were trained by encoding the data ($\mathbf{x}$) and the approximate posterior gradients ($\nabla_\lambda \mathcal{L}$) for 5 inference iterations. All models were trained for 1,500 epochs.

## C.4 ADDITIONAL INFERENCE ITERATIONS

The model architecture for all encoding schemes was identical to that used in the previous section. All models were trained by evaluating the ELBO with a single approximate posterior sample. We trained all models for 1,500 epochs. We were unable to run multiple trials for each experimental set-up, but on a subset of runs for standard and iterative inference models, we observed that final performance had a standard deviation less than 0.1 nats, below the difference in performance between models trained with different numbers of inference iterations.

## C.5 COMPARISON WITH STANDARD INFERENCE MODELS

Directly comparing inference optimization performance between inference techniques is difficult; inference estimates affect learning, resulting in models that are better suited to the inference scheme. Instead, to quantitatively compare the performance between standard and iterative inference models, we trained models with the same architecture using each inference model form. We trained both one-level and hierarchical models on MNIST and one-level models on CIFAR-10. In each case, iterative inference models were trained by encoding the data and errors for 5 inference iterations. We estimated marginal log-likelihoods for each model using 5,000 importance weighted samples per data example.

### C.5.1 MNIST

For MNIST, one-level models consisted of a latent variable of size 64, and the inference and generative networks both consisted of 2 hidden layers, each with 512 units. Hierarchical models consisted of 2 levels with latent variables of size 64 and 32 in hierarchically ascending order. At each level, the inference and generative networks consisted of 2 hidden layers, with 512 units at the first level and 256 units at the second level. At the first level of latent variables, we also used a set of deterministic

units, also of size 64, in both the inference and generative networks. Hierarchical models included batch normalization layers at each hidden layer of the inference and generative networks; we found this beneficial for training both standard and iterative inference models. Both encoder and decoder networks in the hierarchical model utilized highway skip connections at each layer at both levels.

### C.5.2 CIFAR-10

For CIFAR-10, models consisted of a latent variable of size 1024, an encoder network with 3 hidden layers of 2048 units with highway connections, and a decoder network with 1 hidden layer with 2048 units. The variance of the output Gaussian distribution was a global variable for this model. We note that the results reported in table 1 are significantly worse than those typically reported in the literature, however these results are for relatively small fully-connected networks rather than larger convolutional networks. We also experimented with hierarchical iterative inference models on CIFAR-10, but found these models more difficult to train without running into numerical instabilities.

### C.6 COMPARISON WITH VARIATIONAL EM

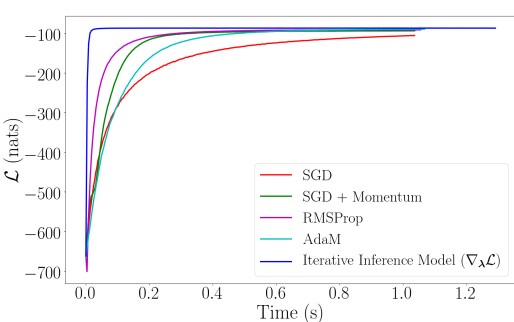

Figure 10: Comparison of inference optimization performance on MNIST test set between iterative inference models and conventional optimization techniques. Performances is plotted in terms of wall-clock time. Iterative inference models still outperform other techniques.

Variational EM is not typically used in practice, as it does not scale well with large models or large data sets. However, because iterative inference models iteratively optimize the approximate posterior parameters, we felt it would be beneficial to provide a comparison of inference optimization performance between iterative inference models and expectation steps from variational EM. We used one-level latent Gaussian models trained with iterative inference models on MNIST for 16 iterations. We compared against vanilla SGD, SGD with momentum, RMSProp, and AdaM, trying learning rates in $\{0.5, 0.4, 0.3, 0.2, 0.1, 0.01, 0.001\}$. In all comparisons, we found that iterative inference models outperformed conventional optimization techniques by large margins. Figure 6 shows the optimization performance on the test set for all optimizers and an iterative inference model trained by encoding the approximate posterior gradients. The iterative inference model quickly arrives at a stable approximate posterior estimate, outperforming all optimizers. It is important to note that the iterative inference model here actually has *less* derivative information than the adaptive optimizers; it only has access to the local gradient. Also, despite only being trained using 16 iterations, the iterative inference remains stable for hundreds of iterations. We also compared the optimization techniques on the basis of wall clock time: Figure 10 reproduces the results from figure 6. We observe that, despite requiring more time per inference iteration, the iterative inference model still outperforms the conventional optimization techniques.

## D EVALUATION ON SPARSE DATA

Concurrent with our work, Krishnan et al. (2017) propose closing the amortization gap by performing inference optimization steps after initially encoding the data with a standard inference model, reporting substantial gains on *sparse, high-dimensional* data, such as text and ratings. We observe similar findings and present a confirmatory experimental result on the RCV1 data set (Lewis et al. (2004)), which consists of 10,000 dimensions containing word counts.

We follow the same processing procedure as Krishnan et al. (2017), encoding data using normalized TF-IDF features and modeling the data using a multinomial distribution. For encoder and decoder, we use 2-layer networks, each with 512 units and ELU non-linearities. We use a latent variable of size 512 as well. The iterative inference model was trained by encoding gradients for 16 steps. We

Table 2: Upper bound on test perplexity on RCV1 for standard and iterative inference models.

| | Perplexity $\approx$ |
|---|---|
| **RCV1** | |
| Standard (VAE) | 385 |
| Iterative | **382** |

evaluate the models by reporting (an upper bound on) perplexity on the test set (Table 2). Perplexity, $P$, is defined as

$$P \equiv \exp(-\frac{1}{N} \sum_i \frac{1}{N_i} \log p(\mathbf{x}_i)), \qquad (63)$$

where $N$ is the number of examples and $N_i$ is the total number of word counts in example $i$. We evaluate perplexity by estimating each $\log p(\mathbf{x}_i)$ with 5,000 importance weighted samples. We observe that iterative inference models outperform standard inference models on this data set by a similar margin reported by Krishnan et al. (2017). Note, however, that iterative inference models here have substantially fewer input parameters than standard inference models (2,048 vs. 10,000). We also run a single optimization procedure for an order of magnitude fewer steps than that of Krishnan et al. (2017).

In Figure 11, we further illustrate the optimization capabilities of the iterative inference model used here. Plotting the average gradient magnitude of the approximate posterior for inference iterations in Figure 11a, we see that over successive iterations, the magnitude decreases. This implies that the model is capable of arriving at near-optimal estimates, where the gradient is close to zero. In Figure 11b, we plot the average relative improvement in the ELBO over inference iterations. We see that the model is quickly able to improve its inference estimates, eventually reaching a relative improvement of roughly 25%.

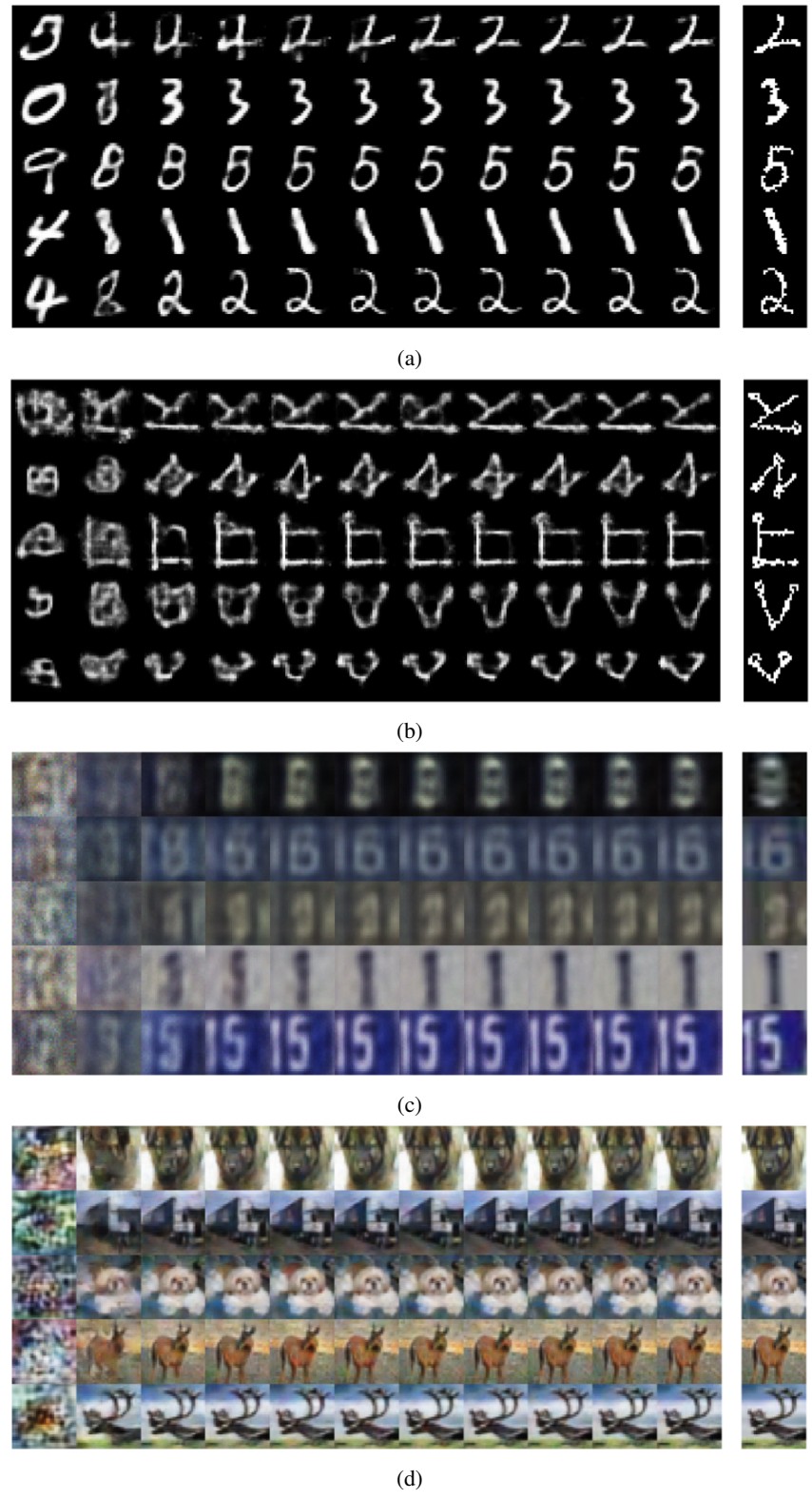

Figure 9: Additional reconstructions over inference iterations (left to right) for test examples from **(a)** MNIST, **(b)** Omniglot, **(c)** Street View House Numbers, and **(d)** CIFAR-10. Corresponding data examples are shown on the far right of each panel.

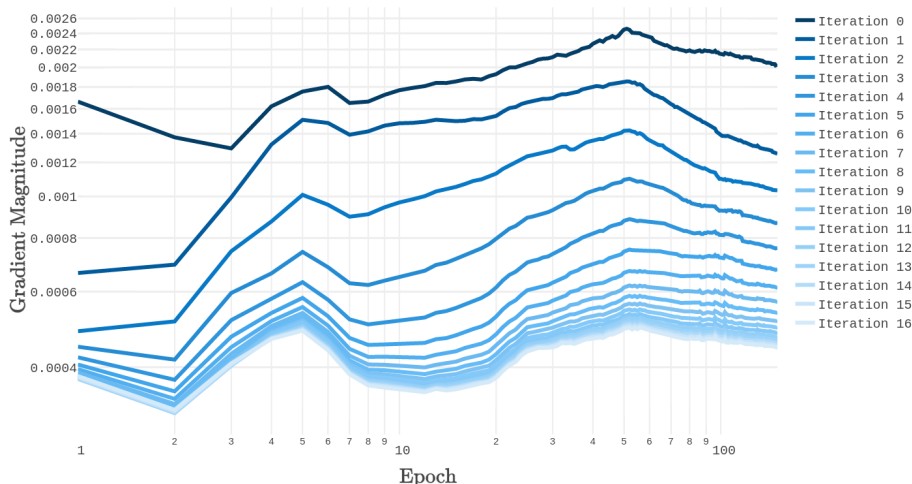

(a) Average gradient magnitudes for the approximate posterior mean.

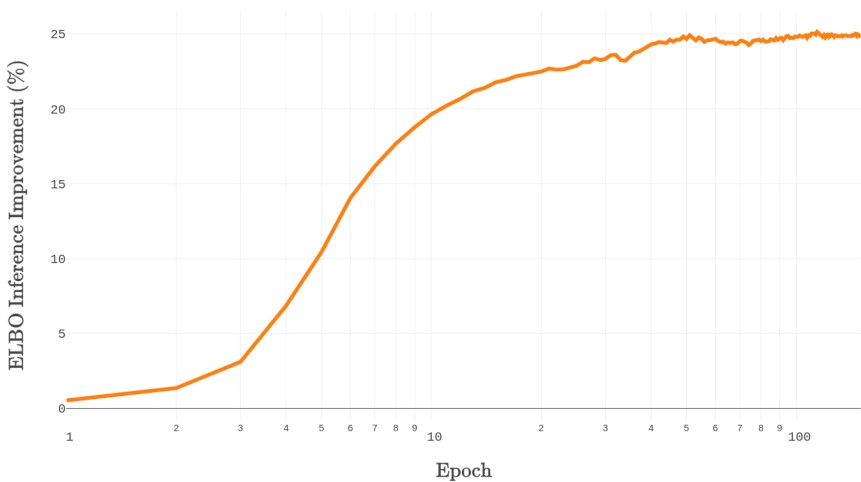

(b) Average relative improvement in evidence lower bound (ELBO) during inference.

Figure 11: Gradient magnitudes and ELBO inference improvement for an iterative inference model trained on the RCV1 data set. (a) The gradient magnitudes for the approximate posterior mean decrease over inference iterations, signifying reaching near-optimal approximate posterior estimates. (b) The iterative inference model immediately learns to start improving its estimates, eventually reaching an average relative improvement of roughly 25% during 16 inference iterations.

