# OpenReview forum: "Learning to Infer"
_ICLR.cc/2018/Conference — Invite to Workshop Track_

### Official Review · AnonReviewer1 · 2017-11-27
**Interesting approach, but the advantages of the approach are unclear**

**Rating:** 5
**Confidence:** 4

**Review:**

This paper proposes an iterative inference scheme for latent variable models that use inference networks. Instead of using a fixed-form inference network, the paper proposes to use the learning to learn approach of Andrychowicz et. al. The parameter of the inference network is still a fixed quantity but the function mapping is based on a deep network (e.g. it could be RNN but the experiments uses a feed-forward network).

My main issue with the paper is that it does not do a good job justifying the main advantages of the proposed approach. It appears that the iterative method should result in "direct improvement with additional samples and inference iterations". I am supposing this is at the test time. It is not clear exactly when this will be useful.

I believe an iterative approach is also possible to perform with the standard VAE, e.g., by bootstrapping over the input data and then using the iterative scheme of Rezende et. al. 2014 (they used this method to perform data imputation).

The paper should also discuss the additional difficulty that arises when training the proposed model and compare them to training of standard inference networks in VAE.

In summary, the paper needs to do a better job in justifying the advantages obtained by the proposed method.

---

> ### Author Response · Authors · 2017-12-16
> **Response to Reviewer 1**
>
> Thank you for your feedback. We hope to clarify points that were unclear through this reply as well as revisions to the paper.
>
> Regarding the utility of our method:
> ``It appears that the iterative method should result in "direct improvement with additional samples and inference iterations"... It is not clear exactly when this will be useful…the paper needs to do a better job in justifying the advantages obtained by the proposed method."
>
> Additional samples and inference iterations help at both training and test time. We presented these experiments to show two aspects of iterative inference models that are distinct from standard inference models, helping readers to distinguish between these models. The main advantage of iterative inference models is that they outperform similar standard inference models in terms of log likelihood, i.e. iterative inference models are better able to capture the data distribution. Increasing the number of samples or inference iterations provides two additional knobs with which to widen this performance gap. We will attempt to make this clearer in the revised paper.
>
> Regarding iterative approaches with VAEs:
> ``I believe an iterative approach is also possible to perform with the standard VAE, e.g., by bootstrapping over the input data and then using the iterative scheme of Rezende et. al. 2014 (they used this method to perform data imputation)."
>
> Such an approach would be qualitatively different than the approach presented here. The data imputation scheme from in (Rezende et. al. 2014) involves iteratively encoding partial observations or reconstructions. If we understand your comment, at best, that approach could only perform as well as a VAE with full observations. Encoding reconstructions would likely introduce further errors.
>
> Regarding training difficulty:
> ``The paper should also discuss the additional difficulty that arises when training the proposed model and compare them to training of standard inference networks in VAE."
>
> We found training iterative inference models to be relatively straightforward and easy to implement. There were no tricks necessary to train these models, and we found that iterative inference models start learning to improve their inference estimates almost immediately. We will include further discussion of this point in Appendix B to assure readers. We will also release code upon publication.

---

> > ### Comment · AnonReviewer1 · 2018-01-11
> > **Revision?**
> >
> > Is there a revision of the paper available? I am assuming there is none because I don't see it in this page.
> >
> > After reading the rebuttal and other reviews, I think that the paper needs plenty of work on clarifying the writing, and as I said in my review, to clarify (and show) the advantages of the proposed method. For the current version, my opinion has not changed (although I have gained clarify about the work and I do think that this work could make an interesting paper).

---

> > > ### Author Response · Authors · 2018-01-12
> > > **Revision and advantages of our method**
> > >
> > > Thank you for your interest in our submission. As it happens, we are currently finishing up a revised version of the paper, which we intend to upload this coming weekend. We hope you will look at the revised paper, as it will include additional clarifications on the points raised by you and the other reviewers.
> > >
> > > With regards to “the advantages of the proposed method,” we seem to misunderstand your comment. We have shown that iterative inference models consistently outperform comparable standard inference models in terms of log-likelihood performance. In other words, iterative inference models result in generative models that are better at fitting data distributions. This is, in and of itself, an advantage of our method. And our experiments on increasing the number of samples and inference iterations demonstrate that we even have the ability to enlarge this advantage. Furthermore, we have shown that iterative inference models converge to similar approximate inference estimates far faster than traditional optimization-based methods. Iterative inference models are therefore more computationally efficient than these methods. This is another clear advantage of our method. As these baselines are the primary methods by which deep latent variable models are currently trained, our work provides the community with an improved method for generative modeling of data.
> > >
> > > We hope you find the revised version of the paper expresses these points more clearly. Please let us know if you have any further comments or questions.

---

> > > > ### Comment · AnonReviewer1 · 2018-01-22
> > > > **Revised version is better. Some more work will improve the impact of the work.**
> > > >
> > > > Thanks for the revised version. I think Figure 8 helps to clarify the contribution a bit more. I think adding a caption and clearly explaining and how phi is obtain from the gradient would be useful.
> > > >
> > > > It is important to propose and compare several 'other' ways of connecting the gradient with the inference network. This will help to understand why the proposed method is a good way to do so? Also, what kind of properties we would want in an inference optimizer to be able to improve over Variational EM as well as VAE. Currently, paper proposes one method but does not add much to the understanding on what kind of methods will generally lead to an improvement over traditional inference methods. In my opinion, if done well, this will help the community move forward.

---

> > > > > ### Author Response · Authors · 2018-01-22
> > > > > **Response to Reviewer 1's Comment**
> > > > >
> > > > > We’re glad that you find the revised version is an improvement and more clearly conveys the contributions of the paper.
> > > > >
> > > > > Iterative inference model parameter gradients are obtained using the reparameterization trick, as with standard inference models. The difference is that these gradients are obtained and averaged over inference iterations. We will clarify this point in the caption of Figure 8.
> > > > >
> > > > > We are unclear what is meant by proposing and comparing ‘other’ ways of connecting the gradient with the inference network. We followed the method of Andrychowicz et al. for inputting the gradient, using the sign and log of the gradient. If the reviewer means that exploring other methods of processing the gradient may be useful, then we agree, but this does not impact the main contribution of this paper: one can learn to infer using gradients. We hope to further explore this technical detail for the final version of the paper.
> > > > >
> > > > > As far as desirable properties for an inference procedure, it is a speed accuracy trade-off. We want a model that is capable of arriving at near-optimal inference estimates in a reasonable amount of time. We have demonstrated that iterative inference models outperform standard inference models in terms of accuracy, achieving similar performance as variational EM in a fraction of the time. We have additionally shown that encoding errors and/or the data can arrive at similar or improved estimates even faster. Please let us know if this point is unclear in the paper so that we can clarify it further.

---

### Official Review · AnonReviewer3 · 2017-11-28
**A nice application of learning-to-learn to address some limitations of purely feedforward amortized inference in VAEs.**

**Rating:** 6
**Confidence:** 5

**Review:**

This paper proposes a learning-to-learn approach to training inference networks in VAEs that make explicit use of the gradient of the log-likelihood with respect to the latent variables to iteratively optimize the variational distribution. The basic approach follows Andrychowicz et al. (2016), but there are some extra considerations in the context of learning an inference algorithm.

This approach can significantly reduce the amount of slack in the variational bound due to a too-weak inference network (above and beyond the limitations imposed by the variational family). This source of error is often ignored in the literature, although there are some exceptions that may be worth mentioning:
* Hjelm et al. (2015; https://arxiv.org/pdf/1511.06382.pdf) observe it for directed belief networks (admittedly a different model class).
* The ladder VAE paper by Sonderby et al. (2016, https://arxiv.org/pdf/1602.02282.pdf) uses an architecture that reduces the work that the encoder network needs to do, without increasing the expressiveness of the variational approximation.
* The structured VAE paper by Johnson et al. (2016, https://arxiv.org/abs/1603.06277) also proposes an architecture that reduces the load on the inference network.
* A very recent paper by Krishnan et al. (https://arxiv.org/pdf/1710.06085.pdf, posted to arXiv days before the ICLR deadline) is probably closest; it also examines using iterative optimization (but no learning-to-learn) to improve training of VAEs. They remark that the benefits on binarized MNIST are pretty minimal compared to the benefits on sparse, high-dimensional data like text and recommendations; this suggests that the learning-to-learn approach in this paper may shine more if applied to non-image datasets and larger numbers of latent variables.

I think this is good and potentially important work, although I do have some questions/concerns about the results in Table 1 (see below).


Some more specific comments:

Figure 2: I think this might be clearer if you unrolled a couple of iterations in (a) and (c).

(Dempster et al. 1977) is not the best reference for this section; that paper only considers the case where the E and M steps can be done in closed form on the whole dataset. A more relevant reference would be Stochastic Variational Inference by Hoffman et al. (2013), which proposes using iterative optimization of variational parameters in the inner loop of a stochastic optimization algorithm.

Section 4: The statement p(z)=N(z;mu_p,Sigma_p) doesn’t quite match the formulation of Rezende&Mohamed (2014). First, in the case where there is only one layer of latent variables, there is almost never any reason to use anything but a normal(0, I) prior, since the first weight matrix of the decoder can reproduce the effects of any mean or covariance. Second, in the case where there are two or more layers, the joint distribution of all z need not be Gaussian (or even unimodal) since the means and variances at layer n can depend nonlinearly on the value of z at layer n+1. An added bonus of eliminating the mu_p, Sigma_p: you could get rid of one subscript in mu_q and sigma_q, which would reduce notational clutter.

Why not have mu_{q,t+1} depend on sigma_{q,t} as well as mu_{q,t}?

Table 1: These results are strange in a few ways:
* The gap between the standard and iterative inference network seems very small (0.3 nats at most). This is much smaller than the gap in Figure 5(a).
* The MNIST results are suspiciously good overall, given that it’s ultimately a Gaussian approximation and simple fully connected architecture. I’ve read a lot of papers evaluating that sort of model/variational distribution as a baseline, and I don’t think I’ve ever seen a number better than ~87 nats.

---

> ### Author Response · Authors · 2017-12-16
> **Response to Reviewer 3**
>
> Thank you for your feedback. We hope to clarify points that were unclear through this reply as well as revisions to the paper.
>
> ``A very recent paper by Krishnan et al. (https://arxiv.org/pdf/1710.06085.pdf, posted to arXiv days before the ICLR deadline) is probably closest; it also examines using iterative optimization (but no learning-to-learn) to improve training of VAEs. They remark that the benefits on binarized MNIST are pretty minimal compared to the benefits on sparse, high-dimensional data like text and recommendations; this suggests that the learning-to-learn approach in this paper may shine more if applied to non-image datasets and larger numbers of latent variables."
>
> We became aware of the work by Krishnan et al. after the deadline, and we will cite them as concurrent work. We find it interesting that they did not see a larger improvement on binarized MNIST, as this may point to qualitative differences between their approach and learned optimization. We plan to include additional experiments in the appendix applying iterative inference models to sparse data.
>
> ``Figure 2: I think this might be clearer if you unrolled a couple of iterations in (a) and (c)."
>
> Thank you for the suggestion. We plan to include an additional figure in the appendix showing these iterative approaches unrolled in time.
>
> ``(Dempster et al. 1977) is not the best reference for this section; that paper only considers the case where the E and M steps can be done in closed form on the whole dataset. A more relevant reference would be Stochastic Variational Inference by Hoffman et al. (2013)..."
>
> We will cite Hoffman et al. (2013). We were initially hesitant to cite this reference as they make use of natural gradients, which are absent in this work.
>
> `` The statement p(z)=N(z;mu_p,Sigma_p) doesn’t quite match the formulation of Rezende&Mohamed (2014…in the case where there are two or more layers, the joint distribution of all z need not be Gaussian (or even unimodal)…"
>
> We chose this formulation in the derivation because it provides a more general treatment. As pointed out, it is unnecessary in the case of a one-level model. However, this formulation is applicable in the hierarchical case, where the prior is typically some arbitrary factorized Gaussian density. The discussion in Section 4 applies to one-level models, which are most commonly used in practice. You are correct that a hierarchical prior need not take the form of a Gaussian, and we discuss this model form in further detail in Appendix A.6. We will attempt to make this point clearer.
>
> ``Why not have mu_{q,t+1} depend on sigma_{q,t} as well as mu_{q,t}?"
>
> This is, in fact, what we do in practice. VAEs have typically been presented as having separate functions for each approximate posterior term, which then share parameters to simplify the model and make learning more efficient. We followed this convention.
>
> ``The gap between the standard and iterative inference network seems very small (0.3 nats at most). This is much smaller than the gap in Figure 5(a)."
>
> Table 1 presents negative log-likelihood estimates using 5,000 importance weighted samples, whereas all other figures show lower bound estimates using a single sample. The gap between negative log-likelihood estimates and lower bound estimates need not be the same, as they depend on the tightness of the bounds. We will make this distinction clearer in the paper.
>
> ``The MNIST results are suspiciously good overall...I don’t think I’ve ever seen a number better than ~87 nats."
>
> Our results agree with (Sønderby et al., 2016), who report a NLL of ~85 nats for a nearly identical model architecture (compare with our ~84 nats for a standard inference model). As in their experiments, we use the dynamically binarized version of MNIST, which results in higher log-likelihoods as compared with statically binarized MNIST. The additional ~1 nat gap is likely due to different activation functions and encoding architecture. We used exponential linear units (ELU), which we have always found to yield superior performance over leaky ReLUs used in (Sønderby et al., 2016). We also used residual encoding networks, which tend to perform better.

---

> > ### Comment · AnonReviewer3 · 2018-01-13
> > **Thanks for the clarifications**
> >
> > I look forward to the revised version.

---

### Official Review · AnonReviewer2 · 2017-11-28
**Learning-to-learn is applied to the inference step in VAE models to improve speed and accuracy**

**Rating:** 5
**Confidence:** 4

**Review:**

Instead of either optimization-based variational EM or an amortized inference scheme implemented via a neural network as in standard VAE models, this paper proposes a hybrid approach that essentially combines the two.  In particular, the VAE inference step, i.e., estimation of q(z|x), is conducted via application of a recent learning-to-learn paradigm (Andrychowicz et al., 2016), whereby direct gradient ascent on the ELBO criteria with respect to moments of q(z|x) is replaced with a neural network that iteratively outputs new parameter estimates using these gradients.  The resulting iterative inference framework is applied to a couple of small datasets and shown to produce both faster convergence and a better likelihood estimate.

Although probably difficult for someone to understand that is not already familiar with VAE models, I felt that this paper was nonetheless clear and well-presented, with a fair amount of useful background information and context.  From a novelty standpoint though, the paper is not especially strong given that it represents a fairly straightforward application of (Andrychowicz et al., 2016).  Indeed the paper perhaps anticipates this perspective and preemptively offers that "variational inference is a qualitatively different optimization problem" than that considered in (Andrychowicz et al., 2016), and also that non-recurrent optimization models are being used for the inference task, unlike prior work.  But to me, these are rather minor differentiating factors, since learning-to-learn is a quite general concept already, and the exact model structure is not the key novel ingredient.  That being said, the present use for variational inference nonetheless seems like a nice application, and the paper presents some useful insights such as Section 4.1 about approximating posterior gradients.

Beyond background and model development, the paper presents a few experiments comparing the proposed iterative inference scheme against both variational EM, and pure amortized inference as in the original, standard VAE.  While these results are enlightening, most of the conclusions are not entirely unexpected.  For example, given that the model is directly trained with the iterative inference criteria in place, the reconstructions from Fig. 4 seem like exactly what we would anticipate, with the last iteration producing the best result.  It would certainly seem strange if this were not the case.  And there is no demonstration of reconstruction quality relative to existing models, which could be helpful for evaluating relative performance.  Likewise for Fig. 6, where faster convergence over traditional first-order methods is demonstrated; but again, these results are entirely expected as this phenomena has already been well-documented in (Andrychowicz et al., 2016).

In terms of Fig. 5(b) and Table 1, the proposed approach does produce significantly better values of the ELBO critera; however, is this really an apples-to-apples comparison?  For example, does the standard VAE have the same number of parameters/degrees-of-freedom as the iterative inference model, or might eq. (4) involve fewer parameters than eq. (5) since there are fewer inputs?  Overall, I wonder whether iterative inference is better than standard inference with eq. (4), or whether the recurrent structure from eq. (5) just happens to implicitly create a better neural network architecture for the few examples under consideration.  In other words, if one plays around with the standard inference architecture a bit, perhaps similar results could be obtained.


Other minor comment:
* In Fig. 5(a), it seems like the performance of the standard inference model is still improving but the iterative inference model has mostly saturated.
* A downside of the iterative inference model not discussed in the paper is that it requires computing gradients of the objective even at test time, whereas the standard VAE model would not.

---

> ### Author Response · Authors · 2017-12-16
> **Response to Reviewer 2**
>
> Thank you for your feedback. We hope to clarify points that were unclear through this reply as well as revisions to the paper.
>
> Regarding novelty:
> ``…the paper…represents a fairly straightforward application of (Andrychowicz et al., 2016). …learning-to-learn is a quite general concept already, and the exact model structure is not the key novel ingredient."
>
> While our work is related to that of (Andrychowicz et al., 2016), there are several novel distinctions:
> 1.	we apply learned optimization models to variational inference,
> 2.	we empirically demonstrate that feedforward networks can perform optimization, whereas previous works required recurrent networks,
> 3.	we develop a novel encoding form that approximates derivatives.
> To the best of our knowledge, these findings are not fully discussed or demonstrated in the literature.
> Unlike learning, variational inference optimization operates over fewer steps and is performed separately for each example, rather than across different tasks. Furthermore, our experiments with hierarchical latent variable models demonstrate a qualitatively different form of optimization model, split across separate networks on multiple levels of optimized variables.
> The optimization model architecture is an important contribution, as all previous works have only used recurrent neural networks, implicitly assuming that learned optimization requires coordination over multiple steps. We have shown that feedforward networks can learn to perform optimization, outperforming optimizers like ADAM and RMSProp that capture additional curvature information from decaying moments.
> The reviewer states, “the paper presents some useful insights such as Section 4.1 about approximating posterior gradients.” We have shown that computing approximate posterior gradients is unnecessary; a model can learn to optimize using locally computed errors. To the best of our knowledge, this is the first time this observation has been explicitly identified in the literature, providing a novel form of learned optimization models.
>
> Regarding seemingly unsurprising results:
> ``...most of the conclusions are not entirely unexpected."
> ``...these results are entirely expected as this phenomena has already been well-documented in (Andrychowicz et al., 2016)."
>
> The results on inference optimization capabilities (Section 5.1 and Figure 6) are interesting for the reason that they are what we would expect. It’s un-intuitive and surprising that an iterative inference model can learn to optimize a generative model, and our results verify that this is done in a reasonable manner. Few works in the VAE literature have discussed optimization performance, so it is instructive to visualize and quantify how various methods compare.
>
> Regarding experimental comparisons:
> ``...is this really an apples-to-apples comparison?…might eq. (4) involve fewer parameters than eq. (5) since there are fewer inputs?...if one plays around with the standard inference architecture a bit, perhaps similar results could be obtained."
>
> The gradient encoding iterative inference model (eq. 5) in Figure 5b has fewer input parameters (256 vs. 784), yet outperforms the standard inference model. We will clarify this point. As the reviewer points out, a perfect comparison of models is difficult to perform: varying numbers of inputs result in varying numbers of input parameters. Yet, the number of parameters processing information from the data is constant across both models, showing that gradients and errors contain additional information. Regarding our results, we found that iterative inference models outperformed standard models across a variety of architectures (varying network/latent width, residual/dense connections, batch norm, etc.) on the benchmark data sets. The experiments are representative of this finding, which we hope to clarify in the revised paper.
>
> Miscellaneous:
> ``A downside…is that it requires computing gradients of the objective even at test time..."
>
> Iterative inference models that encode gradients require these gradients at test time, which we will state more clearly. However, the error encoding models that we introduce do not require these gradients, one of their benefits that we highlight.
>
> ``…there is no demonstration of reconstruction quality relative to existing models, which could be helpful for evaluating relative performance."
>
> The purpose of Figure 4 is to provide a qualitative verification of our inference optimization, not to demonstrate superior reconstruction quality. It would also be difficult for humans to visually inspect these differences, as they likely involve slight differences in pixel intensities.
>
> ``In Fig. 5(a), …the standard inference model is still improving but the iterative inference model has mostly saturated."
>
> We agree, but this does not impact the main empirical findings from section 5.2: iterative inference models improve significantly with more approximate posterior samples.

---

### Author Response · Authors · 2018-01-15
**Revision Added**

We have uploaded a revised version of the submission, which attempts to take the reviewers’ comments into account. Again, we thank the reviewers for their help with this process. We specifically highlight the following additions:

- Additional empirical results on the sparse, high-dimensional RCV1 text data set (Appendix D), following Krishnan, et al. Using a multinomial output, we also observe empirical benefits using iterative inference models over standard inference models. We also include Figure 11, which further illustrates learned inference optimization on this data set.
- Figure 8, which shows unrolled computational graphs for each inference scheme. We hope this helps in clarifying each process.
- Clarification of distinctions / novel aspects of this work over previous methods at the end of Section 3.1.
- Clarification of the relative number of input parameters in each model in Section 5.2.
- Discussion of difficulty of training iterative inference models in Appendix B.4.
- Clarification on where we report ELBO values and NLL values in Section 5 (first paragraph).
- Additional sentences in Section 3.1 (2nd and 3rd paragraphs) discussing the amortization gap, i.e. the gap in performance by assuming an amortized inference scheme.
- Citations for Hoffman et al., 2013; Krishnan et al., 2017; Cremer et al., 2017.

Finally, we would like to close by stating that we feel the content of this submission provides many useful insights to the larger Bayesian deep learning community. We have taken the VAE, one of the most popular models in this area, and provided a novel method by which to perform inference optimization. While the idea of iterative inference optimization may initially seem counterintuitive, we have empirically demonstrated that moving beyond the typical data-encoding paradigm has clear advantages in terms of modeling performance. We have demonstrated the feasibility and success of our method on multiple data sets using various output modeling distributions. This work provides a more detailed view of inference optimization and will hopefully enable further work in amortized variational inference and learned optimization.

---

### Decision · Program_Chairs · 2018-01-29
**ICLR 2018 Conference Acceptance Decision**

**Decision:**

Invite to Workshop Track

**Comment:**

This paper is intersting but has a few flaws that still need to be addressed. As one reviewer noted, "the authors seems to have simply applied the method of Andrychowicz et al. If they added some discussion and experiments clearly showing why this is a better way to improve the existing inference methods, the paper might have more impact.".
Overall, this work builds on existing work, but does not really dig deep enough for answers to these questions raised by the reviewers. The committee still feels this paper will be of great value at ICLR and recommends it for a workshop paper.